# *i*Finder: Structured Zero-Shot Vision-Based LLM Grounding for Dash-Cam Video Reasoning

**Manyi Yao**[†]**, Bingbing Zhuang**[‡]**, Sparsh Garg**[‡]**, Amit Roy-Chowdhury**[†],
**Christian Shelton**[†]**, Manmohan Chandraker**[‡,⋆]**, Abhishek Aich**[‡]
[‡]NEC Laboratories, America, [†]University of California, Riverside,
[⋆]University of California, San Diego

## Abstract

Grounding large language models (LLMs) in domain-specific tasks like post-hoc dash-cam driving video analysis is challenging due to their general-purpose training and lack of structured inductive biases. As vision is often the sole modality available for such analysis (*i.e.* no LiDAR, GPS, *etc.*), existing video-based vision-language models (V-VLMs) struggle with spatial reasoning, causal inference, and explainability of events in the input video. To this end, we introduce *i***Finder**, a structured semantic grounding framework that decouples perception from reasoning by translating dash-cam videos into a hierarchical, interpretable data structure for LLMs. *i***Finder** operates as a modular, training-free pipeline that employs pretrained vision models to extract critical cues—object pose, lane positions, and object trajectories—which are hierarchically organized into frame- and video-level structures. Combined with a three-block prompting strategy, it enables step-wise, grounded reasoning for the LLM to refine a peer V-VLM's outputs and provide accurate reasoning. Evaluations on four public dash-cam video benchmarks show that *i***Finder**'s proposed grounding with domain-specific cues—especially object orientation and global context—significantly outperforms end-to-end V-VLMs on four zero-shot driving benchmarks, with up to 39% gains in accident reasoning accuracy. By grounding LLMs with driving domain-specific representations, *i***Finder** offers a zero-shot, interpretable, and reliable alternative to end-to-end V-VLMs for post-hoc driving video understanding.

## 1 Introduction

Grounding large language models (LLMs) to domain-specific requirements remains a significant challenge due to their general-purpose training and lack of inductive bias toward structured domain knowledge [1]. LLMs are often pretrained on broad internet data, which may lead to imprecise outputs when applied to specialized domains such as driving video understanding [2]. Furthermore, the lack of interpretability and limited mechanisms for precise reasoning make it difficult to guarantee reliability in high-stakes applications [3]. In this paper, we tackle the problem of generating causally and spatially grounded LLM responses to user queries, given a front-view dash-cam video. The objective is to produce explanations that accurately reflect the underlying events in the scene.

The proliferation of advanced driver assistance systems has led to an abundance of video data, often serving as the primary or sole modality for system analysis and validation. In many consumer-grade and fleet-level applications, supplementary sensors like LiDAR, GPS, or CAN bus data are either unavailable or impractical to collect due to cost and integration complexities [4, 5]. Consequently, extracting meaningful insights from visual spatio-temporal data becomes crucial for assessing system responses, diagnosing failures, and refining vehicle performance [6–11]. However, developing video analysis systems that generate grounded and reliable responses *solely* from camera input remains a formidable challenge, without external supervision [12, 13].

39th Conference on Neural Information Processing Systems (NeurIPS 2025).

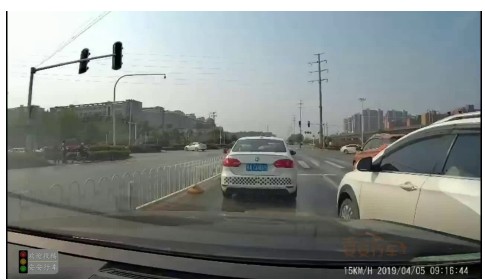

**Q:** *What is the cause of the accident?*
**Ground Truth:** The car does not give way to normal driving vehicles when turning or changing lanes.

**VideoLLaMA2:** There is no accident in the video. However, there are several instances of cars stopping at red lights and waiting for them to turn green before continuing on their way.

**VideoLLaVA:** The cause of the accident is that the car in front of the white car suddenly stops, causing the white car to collide with the car in front of it.

**VideoChat2:** The cause of the accident is the white car's failure to stop at the red traffic light.

**DriveMM:** There is no accident in the video. The traffic is flowing smoothly with vehicles maintaining a safe distance from each other.

**iFinder:** A white car in the right lane attempted to cut into the ego lane, causing a near-collision, as seen in Object ID 13 moving from 9.78m (Frame 24) to 6.14m (Frame 40), with rot_y: -2.1, indicating a leftward turn.

Figure 1: **Advantages of *i*Finder.** Baselines VideoLLaMA2 [14], VideoLLaVA [15], and DriveMM [16] struggle with spatial reasoning, and fine-grained scene understanding, misinterpreting critical cues. *i*Finder's structured scene approach mitigates these errors for more accurate responses.

A seemingly straightforward solution to address the above problem is to develop end-to-end video-based Vision-Language Models (V-VLMs) [14–16]. However, although existing V-VLMs generate reasonable responses, they exhibit limitations in spatial reasoning, causal inference, and fine-grained scene understanding [17–19] as shown in Figure 1. This means misinterpretation of critical visual cues can lead to incorrect conclusions about hazards, traffic signals, or object presence. Furthermore, incorporating new functionality without requiring extensive retraining or fine-tuning is difficult with VLMs [20]. To address these limitations, we build on the principle that perception should be decoupled from LLM reasoning. In particular, we propose *i*Finder, a vision-language pipeline that extracts driving domain-relevant visual cues and passes them to the LLM via structured prompts, enabling post-hoc scenario understanding through symbolic, temporally grounded reasoning. This structured scene representations encode dynamic object pose, orientation, and semantic lane context in a hierarchical format, enabling symbolic reasoning over frame-indexed data. *i*Finder leverages pre-trained vision models to extract domain-relevant cues and organizes them into a hierarchical data format. This structured input enables the LLM to reason accurately about driving scenarios, correcting or augmenting generic V-VLM outputs with grounded, verifiable evidence tailored to the driving video context. For example, in Figure 1, we can observe that unlike baselines [14–16, 21] that provide incorrect or generic explanations, *i*Finder correctly identifies the white car's lane-cutting maneuver, aligning with the ground truth. It leverages the data structure of the input video and uses object tracking ('Object ID 13'), distance change ('9.78m → 6.14m'), and orientation ('rot_y = -2.1 to indicate a left turn') to draw conclusions. Our approach not only mitigates the limitations of end-to-end V-VLMs but also improves reliability and transparency.

The complete *i*Finder pipeline is as follows. The process begins with input video frames that are first undistorted to correct lens distortion. These frames are then processed by a suite of pre-trained vision modules that extract critical driving cues: scene context, ego-vehicle motion, 2D/3D object detections, object tracking, lane assignments, object distances, and semantic attributes. This information is hierarchically structured into video-level (*e.g.*, global context, ego state, peer VLM response) and frame-level (*e.g.*, object properties per frame) representations. Simultaneously, a peer V-VLM provides an initial answer to the user query, which may contain inaccuracies. The final reasoning is handled by the LLM, which is prompted using three components—peer instruction, step-by-step reasoning guidance, and key explanations of the structured inputs. Combining the peer's response with structured, domain-grounded visual cues enables the LLM to produce accurate and interpretable answers for complex driving event scenarios. Through rigorous analysis on four public benchmarks, we provide three new insights to the community:

- **Improved Performance via Structured Grounding.** By decoupled perception and LLM reasoning, and grounding LLMs in hierarchical, interpretable video representations, *i*Finder beats both generalist and driving-specific VLMs, without any fine-tuning. *i*Finder, for example, in the accident reasoning dataset MM-AU [22], beats the best performing generalist V-VLM by 10.5% and driving-specialized V-VLM by 39.17%.
- **Enhanced Explainability through Explicit Reasoning.** Unlike end-to-end VLMs that rely on implicit cues, *i*Finder provides symbolic cues—such as object orientation, lane context, and distance—that allows the LLM to generate transparent, verifiable explanations, as shown in Figure 5.

- **Object-orientation and Global Environmental Dominate.** Surprisingly, our ablation studies in Table 5 reveal that object-orientation and global environmental context contribute more to reliable post-hoc reasoning than other factors like distance and lane location. For instance, removing object orientation information led to a significant drop in reasoning accuracy ($\sim$4.5%), while removing distance or lane location had a comparatively minor effect ($\sim$2.7%).

## 2 Related Works

**Post-hoc Driving Video Analysis.** Post-hoc driving video analysis for front-cam setting (rather than surveillance-cam setting, such as in traffic intersections [23]) has recently become a strong research interest. Prior works on driving video analysis [22, 24] have been dominantly focused on accidents and their possible prevention analysis. Different from these, [25] provides a detailed benchmark in which models can be tested on their understanding of the ego-vehicle's perspective based on their understanding of dynamic scenes, rather than merely their ability to describe the video event. Recent work has explored zero-shot hazard identification and out-of-distribution challenges, with [26] introducing the COOOL benchmark for evaluating autonomous driving models on out-of-label objects, [27] demonstrating zero-shot hazard identification approaches on this benchmark, and [28] proposing multi-agent vision-language systems for detecting novel hazardous objects. In this work, unlike prior driving VLMs [16, 18], we develop a vision-language pipeline that solely focuses on such offline driving video analysis.

**Video Foundational Models for Driving Videos.** The success of the *image*-bases VLSs has sparked a growing interest in the *video*-based VLMs within the research community. In the video-language domain, the prevailing approach now involves large-scale video-text pre-training followed by fine-tuning for specific tasks [15, 21, 29–48]. Within the driving video domain, multiple image-language models have been introduced [49–59] for various driving-related tasks. Although these methods perform well, their effectiveness is restricted to specific scenarios (Bird's-Eye-View-based representations, multi-view representations, etc.) and driving-specific objectives that dominantly do not aid offline video analysis tasks. For video-domain driving specific foundational models, methods like [16, 18, 60] have been introduced. However, they lack the capability to process different driving video with fine-grained analysis like ego-car attributes. Moreover, these VLMs are designed for tasks to represent in-moment vehicle driver, rather than video analysis.

## 3 Method

**Motivation.** We argue that grounding general-purpose LLMs using structured, interpretable scene representations offers a more accurate alternative than V-VLMs. Further, decoupling perception from reasoning enables a more reliable analysis system. Hence, we introduce a modular framework that grounds general-purpose LLMs using structured, interpretable representations derived from pretrained vision modules. These modules extract critical scene attributes—such as object pose, lane semantics, and motion cues—which are hierarchically organized into a domain-specific data structure. This explicit grounding enables the LLM to reason with verifiable, context-aware inputs rather than relying solely on implicit visual cues. Before detailing each module, we summarize the full *i*Finder pipeline: raw dash-cam video frames are first undistorted and processed through pretrained vision models to extract scene-level and object-level cues. These are hierarchically organized into a structured data format and passed—along with a peer V-VLM's output—into a three-block prompt that guides a general-purpose LLM to generate post-hoc reasoning responses.

### 3.1 Proposed Framework

***i*Finder pipeline.** Our pipeline is shown in Figure 2. Let a front-view dash-cam video be represented as a sequence of $T$ frames as $\boldsymbol{V} = (\boldsymbol{I}_{d1}, \boldsymbol{I}_{d2}, \cdots, \boldsymbol{I}_{dT})$, where each frame $\boldsymbol{I}_{dt} \in \mathbb{R}^{H \times W \times 3}$ is the $t$-th *d*istorted or unrectified image of height $H$ and width $W$. We define a unified function $\mathcal{F}$ that maps input video $\boldsymbol{V}$ to structured data $\mathcal{D}$ for grounding the LLM.

$$\mathcal{F} : \underbrace{\mathbb{R}^{T \times H \times W \times 3}}_{\text{space of all front-view dash-cam videos}} \rightarrow \mathcal{D} \tag{1}$$

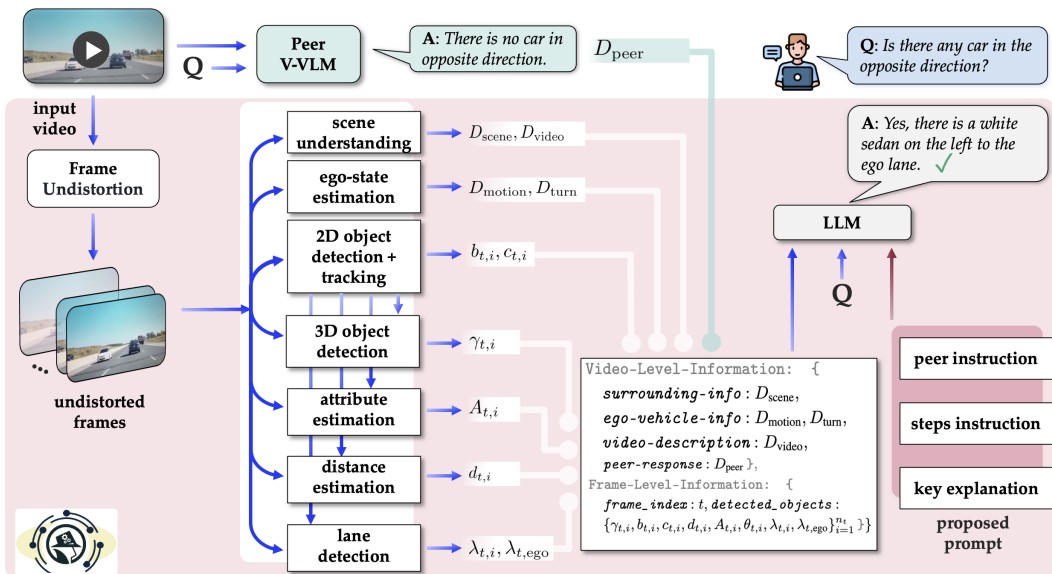

Figure 2: **iFinder overview.** The proposed pipeline transforms key scene properties such as object detection, lane detection, depth estimation, and ego-state estimation, into structured data, which, combined with peer-generated insights, enables the LLM to perform accurate and interpretable driving scenario analysis.

Finally, the structured data $\mathcal{D}$ is passed to an LLM, denoted $\mathcal{F}_{\text{LLM}}$, along with a prompt $P_{\text{LLM}}$, to generate the final response $\boldsymbol{R}$ to the user's query. Next, we describe the function $\mathcal{F}$ that aims to extract $\boldsymbol{D}$ in an LLM-friendly manner. In practice, $\mathcal{F}$ is decomposed into multiple specialized modules: 2D/3D object detection, lane localization, distance estimation, attribute estimation, ego-state estimation, and scene understanding. Each of the modules is instantiated with popular pre-trained open-source models, requiring no fine-tuning when used within **iFinder**'s pipeline.

### 3.1.1 Visual Information Extraction

**Step 1: Undistorting frames.** Front-view dash-cam videos often exhibit distortions due to the characteristics of front-facing cameras [61]. To address this and ensure optimal performance for perception models to follow, **iFinder** begins by correcting these distortions through the estimation of camera intrinsics $\boldsymbol{K}$ (focal length and principal point), and distortion coefficients, including radial distortion $[k_1, k_2, k_3]$ and tangential distortion $[p_1, p_2]$ coefficients. To correct for lens distortion, we compute a mapping function $\mathcal{R}$ that transforms coordinates in the rectified (undistorted) image $\boldsymbol{I}$ to corresponding coordinates in the distorted image $\boldsymbol{I}_d$. The rectified image is obtained by:

$$\boldsymbol{I}(x,y) = \boldsymbol{I}_d(\mathcal{R}^{-1}(x,y)), \tag{2}$$

where $\mathcal{R}^{-1}(x,y)$ computes the corresponding distorted coordinates for each undistorted pixel location $(x,y)$ using the estimated camera intrinsic matrix and distortion coefficients.

**Step 2: Scene understanding.** This step captures general environment details, such as weather conditions, road structure, and whether it is daytime or nighttime. We also extract a caption describing the events in the video.

*Intuition.* High-level scene understanding, including weather, traffic conditions, and time of day, along with video event descriptions, helps the LLM reason about vehicle movements, pedestrian actions, and traffic interactions.

*Method.* To extract this surrounding environment information $D_{\text{scene}}$, we leverage an image-based VLM $\mathcal{F}_{\text{I-VLM}}$ to enable a precise and reliable interpretation of the video. Now, image-based VLMs lack the ability to capture the evolution of events over time. To capture temporal dynamics, a V-VLM $\mathcal{F}_{\text{V-VLM}}$ is used to generate a detailed event description $D_{\text{video}}$.

$$\mathcal{F}_{\text{I-VLM}} : (\boldsymbol{V}, P_I) \rightarrow D_{\text{scene}}, \quad \mathcal{F}_{\text{V-VLM}} : (\boldsymbol{V}, P_V) \rightarrow D_{\text{video}} \tag{3}$$

**Step 3: Ego-vehicle state estimation.** This step captures the ego-vehicle's motion (moving/stopped) and turn (left/right/straight) action.

*Intuition.* Understanding the ego-vehicle's motion is crucial for reasoning in front-view dash-cam videos where the vehicle's perspective defines the driving scene. Given that we are using front-view dash-cam videos, estimating the camera pose will directly provide the ego-vehicle's motion pattern in the input video. However, these numerical pose outputs are not inherently interpretable by an LLM. To bridge this gap, we transform the raw pose data into human-interpretable driving states by estimating the vehicle's turning behavior and motion status.

*Method.* We use a camera-pose estimation model $\mathcal{F}_{\text{cam-pose}}$ to map video frames $V$ to a sequence of translation vectors $\{T_t\}_{t=1}^T$, where $T_t = (X_t, Y_t, Z_t) \in \mathbb{R}^3$ denotes the camera position at time $t$.

*(A) Vehicle turning estimation.* With $(X_t, Z_t) \forall t \in \{1, \cdots, T\}$, we estimate the heading angle of the camera $\Delta\theta_i$ as

$$\Delta\theta_t = \tan^{-1}\left(Z_{t+1} - Z_t/X_{t+1} - X_t\right) - \tan^{-1}\left(Z_t - Z_{t-1}/X_t - X_{t-1}\right), \qquad (4)$$

Next, $\Delta\theta_t$ is used to classify the vehicle's turn into the three categories ($\tau_a$ as a threshold) as

$$D_{\text{turn}} = \text{"Straight" if } |\Delta\theta_t| < \tau_a, \text{ "Right Turn" if } \Delta\theta_t > \tau_a, \text{ else "Left Turn"} \qquad (5)$$

*(B) Vehicle motion estimation.* We compute the vehicle's motion over a temporal window $g$ using

$$s_t = \|T_{t+g} - T_t\|/g, \quad \forall t \in \{1, \cdots, T - g\}, \qquad (6)$$

where $s_t$ denotes the approximate speed at time $t$, used to classify the vehicle's motion state as

$$D_{\text{motion}} = \text{"Stopped" if } s_t < \tau_s, \text{ else "Moving"} \qquad (7)$$

where $\tau_s$ represents the speed threshold for detecting a stopped vehicle. By incorporating both turning and motion status, we structure the ego-vehicle state as

$$\mathcal{F}_{\text{ego}} : \{(I_t, T_t)\}_{t=1}^T \to (D_{\text{motion}}, D_{\text{turn}}). \qquad (8)$$

While video-level cues provide essential global understanding, many critical driving events, such as pedestrian crossings, vehicle interactions, and traffic signal changes, occur at the frame level. To fully comprehend the driving scenario, we extract frame-level information in **Step 4-7**, ensuring that the model can reason about both long-term motion trends and momentary scene dynamics.

**Step 4: 2D Object detection and tracking.** This step captures and tracks the objects in the video.

*Intuition.* To accurately analyze dynamic interactions of objects with the ego-vehicle in a driving scene, it is necessary to not only detect objects in individual frames but also track them over time using a unique identity. Furthermore, this step provides the necessary foundation for identifying attributes of the objects, such as their lane location, distance, and attributes.

*Method.* For each video frame $I_t$, we apply a 2D object detection model $\mathcal{F}_{\text{2D-det}}$ as $\mathcal{F}_{\text{2D-det}} : (I_t) \to \{(b_{t,i}, c_{t,i})\}_{i=1}^{n_t}$. Here, $b_{t,i} = (x_{\min}, y_{\min}, x_{\max}, y_{\max}) \in \mathbb{R}^4$ is the bounding box and $c_{t,i}$ is the class label for $i$th object. $n_t$ is the number of objects detected in frame $t$. Then, we add a tracker on the detections in order to assign a unique ID $\gamma_{t,i}$ to each detected object $B_t = (b_{t,i}, c_{t,i})$ using a multi-object tracking model $\mathcal{F}_{\text{2D-track}}$ as $\mathcal{F}_{\text{2D-track}} : (B_t, K_{t-1}) \to K_t$. Finally, for each video frame $I_t$, this step provides

$$\mathcal{F}_{\text{2D-det-track}} : (I_t) \to \{\gamma_{t,i}, b_{t,i}, c_{t,i}\}_{i=1}^{n_t}. \qquad (9)$$

**Step 5: Object lane location.** This step assigns the lane location of the object in the scene.

*Intuition.* The ego-vehicle's driving decisions are influenced by the lane positions of surrounding objects. Therefore, it is crucial that the data structure $\mathcal{D}$ encodes lane information for each detected object, particularly for vehicles and pedestrians. Similar to **Step 3**, the numerical outputs of lane detection models are not inherently interpretable by an LLM. To bridge this gap, we transform the raw lane marking data into human-interpretable states by estimating the vehicle's lane location.

*Method.* Once the objects are detected from **Step 4**, we perform the lane assignment as follows. We use a lane detection model $\mathcal{F}_{\text{lane}}$ and first obtain the predicted lane markings in each frame $I_t$ as $\mathcal{F}_{\text{lane}}(I_t) : (I_t) \to \{l_{t,j}\}_{j=1}^{m_t}$. Here, $l_{t,j}$ represents the set of $j$-th lane marking coordinates, and $m_t$ is the total number of detected lane markings. Next, we divide the road into $m_t + 1$ number of lane sections formed by the lane markings. Each lane section is now defined as

$$s_{t,k} = \{(x, y) \mid x_{l_{t,j}} \le x \le x_{l_{t,j+1}}, \, y = [y_{\max, L_t}, H]\}, \qquad (10)$$

where $x_{l_{t,k}}$ is the $x$-coordinate of the $k$-th lane marking, $y_{\max, L_t} = \min_j y_{l_{t,j}}$ is the highest point of all lane markings (assuming image coordinates have the origin at the top-left).

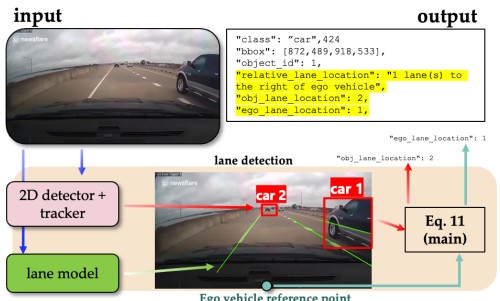

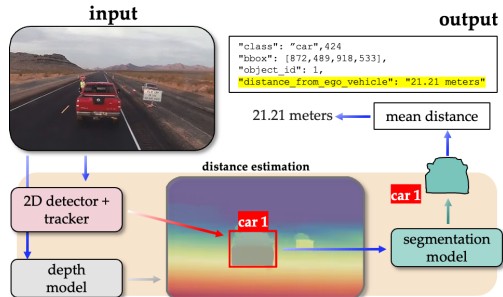

Figure 3: **Lane location estimation.** Detected objects are assigned a lane by mapping the bottom midpoint of the corresponding bounding box (bottom middle point of image for ego) to sections identified by the lane detection model.

Figure 4: **Distance estimation.** Each object's distance is determined by averaging the depth values within its segmented region.

*(A) Object lane estimation.* For each $i$th object in frame $t$, we compute the midpoint $p_{t,i}$ of its bounding box bottom edge as $p_{t,i} = (x_{\min} + x_{\max}/2, y_{\max})$. Then, its lane $\lambda_{t,i}$ is estimated as

$$\lambda_{t,i} = k \quad \text{such that} \quad p_{t,i} \in s_{t,k} . \tag{11}$$

We estimate the ego-vehicle's lane $\lambda_{t,\text{ego}}$ using the bottom-center pixel $p_{t,\text{ego}} = (W/2, H)$ as a reference. The resulting lane data is then added to $\mathcal{D}$ for each frame.

$$\mathcal{F}_{\text{lane}} : (\boldsymbol{I}_t, \boldsymbol{K}_t) \to (\{\lambda_{t,i}\}_{i=1}^{n_t}, \lambda_{t,ego}) . \tag{12}$$

**Step 6:** **Object distance estimation.** This step estimates the distance of the objects in the scene w.r.t. the ego-vehicle from the video.

*Intuition.* Distance-awareness of each object will allow the LLM to analyze collisions, ego-vehicle navigation, and object interaction.

*Method.* Given an input frame $\boldsymbol{I}_t$, a depth estimation model $\mathcal{F}_{\text{depth}}$ predicts a metric depth map $\mathcal{F}_{\text{depth}} : (\boldsymbol{I}_t) \to \boldsymbol{D}_t$ where, $\boldsymbol{D}_t \in \mathbb{R}^{H \times W}$ is the estimated depth map for frame $t$. For $i$th object's bounding box $b_{t,i} = (x_{\min}, y_{\min}, x_{\max}, y_{\max})$, the cropped depth region corresponding to the object is $\boldsymbol{D}_{t,i} = \boldsymbol{D}_t[x_{\min} : x_{\max}, y_{\min} : y_{\max}]$. Next, to make the region of the object more precise and eliminate any background pixel, a segmentation model $\mathcal{F}_{\text{seg}}$ predicts a binary mask $\boldsymbol{M}_{t,i}$ for the object within $b_{t,i}$. The final distance $d_{t,i}$ of the object from the ego-vehicle is computed as the mean distance of the masked region. $d_{t,i} = \text{mean}(\boldsymbol{D}_{t,i} \odot \boldsymbol{M}_{t,i})$, where $\odot$ denotes the element-wise multiplication. Finally, the distance information per object per frame is added to $\mathcal{D}$ as follows.

$$\mathcal{F}_{\text{dist}} : (\boldsymbol{I}_t, \boldsymbol{K}_t) \to \{d_{t,i}\}_{i=1}^{n_t} . \tag{13}$$

A qualitative illustration of **Step 5** and **Step 6** is shown in Figure 3 and Figure 4, respectively.

**Step 7:** **Object attributes.** This step generates object attributes like color to help the LLM distinguish the objects in an interpretable manner.

*Intuition.* Object attributes enhance perception with human-like reasoning, useful for scene interpretation and understanding high-level decision-making.

*Method.* With $b_{t,i} = (x_{\min}, y_{\min}, x_{\max}, y_{\max})$, we use $\mathcal{F}_{\text{I-VLM}}$ to extract object attributes *e.g.*, color of vehicles, traffic light color, *etc.* using prompt $P_d$. See Supplementary Material for details on $P_d$.

$$\mathcal{F}_{\text{I-VLM}} : (\boldsymbol{I}_t[x_{\min} : x_{\max}, y_{\min} : y_{\max}], P_d) \to \{A_{t,i}\}_{i=1}^{n_t} \tag{14}$$

**Step 8:** **3D detection for object orientation.** This step captures the object orientation from the 3D information of objects in the scene w.r.t. the ego-vehicle.

*Intuition.* From **Step 4-8**, all extracted information was from a 2D perspective. However, a critical aspect of the objects missing is their orientation as per the ego-vehicle's view.

*Method.* Using a 3D detection model $\mathcal{F}_{\text{3D-det}}$, we predict $p_t$ 3D bounding boxes and extract the yaw $\theta_{t,i} \in [-\pi, \pi]$ for each object. as $\mathcal{F}_{\text{3D-det}} : (\boldsymbol{I}_t) \to \{\theta_{t,i}\}_{i=1}^{p_t}$. Next, we project each 3D bounding box into 2D image space using the camera intrinsic matrix $\boldsymbol{K}$ from **Step 1**. We then apply the Hungarian algorithm [62] to match projected boxes with detected objects and transfer $\theta_{t,i}$ to the corresponding local object.

### 3.1.2 Proposed Data Structure and Prompt

**Incorporating Peer V-VLM Reasoning.** While structured visual information provides a strong foundation for precise reasoning, we also incorporate a general-purpose V-VLM as a peer module. The peer V-VLM serves two complementary purposes: *(1)* it provides an initial, high-level response to the user query based on raw visual input, and *(2)* it exposes limitations in generic models leading to incorrect explanations. To this end, we query the peer V-VLM using the original video $V$ and extract its response $D_{\text{peer}}$. This response is treated as a first-pass hypothesis, which is later refined by the LLM using structured evidence $\mathcal{D}$. By comparing $D_{\text{peer}}$ with structured scene information, our framework encourages the LLM to ground or correct its reasoning based on verifiable cues. We now describe how the structured data $\mathcal{D}$ is used to guide the final reasoning stage.

**Hierarchical data structure.** $\mathcal{D}$ is designed to organize information in a hierarchical manner, distinguishing between video-level and frame-level details. This structure is particularly intuitive for an LLM because it aligns with how reasoning typically occurs over temporal sequences. Note that the structured representation is provided to the LLM in JSON format.

$$\mathcal{D} = \big\{ \texttt{Video-Level-Information:} \ \{ \qquad\qquad \mathit{response} : D_{\text{peer}} \},$$
$$\mathit{surrounding\text{-}info} : D_{\text{scene}}, \qquad \texttt{Frame-Level-Information:} \ \{$$
$$\mathit{ego\text{-}car\text{-}information} : D_{\text{motion}}, D_{\text{turn}}, \quad \mathit{frame\_index} : t, \mathit{detected\_objects} :$$
$$\mathit{description} : D_{\text{video}}, \qquad\qquad \{\gamma_{t,i}, b_{t,i}, c_{t,i}, d_{t,i}, A_{t,i}, \theta_{t,i}, \lambda_{t,i}, \lambda_{t,\text{ego}}\}_{i=1}^{n_t} \}\}$$

**Three-block prompt.** The prompt $P_{\text{LLM}}$ is designed as a three-block structure to optimize the model's grounded reasoning. The first component, *Key Explanation*, provides a precise and explicit interpretation of the scene representation $\mathcal{D}$, reducing ambiguity in the model's input. Prior work shows that LLMs benefit from well-disambiguated inputs when handling symbolic data [63, 64]. The second component, *Step Instructions*, decomposes the reasoning task into explicit sub-goals. This design is motivated by findings in cognitive science and neural model alignment, where step-by-step prompting improves reasoning accuracy and consistency [65, 66]. The third component, *Peer Instruction*, informs the model that peer-generated answers may be unreliable and explicitly encourages independent reasoning. Together, these three components operationalize input grounding, procedural reasoning, and epistemic caution—three necessary conditions for robust and generalizable performance in tasks involving multi-step inference from structured dynamic scenes. Our prompt has been provided in the Supplementary Material.

**Note on efficiency.** While *iFinder* involves multiple pretrained modules, each step is executed independently and requires no retraining or gradient updates. Inference is parallelizable across modules. Our focus is not on real-time deployment, but on enabling interpretable, post-hoc analysis pipelines—where accuracy is the primary objective.

## 4  Experiments

**Experiment setup.** *iFinder* leverages a combination of state-of-the-art models in each of its steps. In **Step 1**, we use GeoCalib [67] for estimating camera parameters and distortion coefficients. To correct the lens distortion, we use OpenCV's undistort [68] function for $\mathcal{R}$. In **Step 2**, we use InternVL [69] for $\mathcal{F}_{\text{I-VLM}}$ and VideoLLaMA2 [14] for $\mathcal{F}_{\text{V-VLM}}$. In **Step 3**, we use DROID-SLAM [70] for $\mathcal{F}_{\text{cam-pose}}$. In **Step 4** for $\mathcal{F}_{\text{2D-det}}$, we use OWL-V2 [71] and ByteTracker [72] for $\mathcal{F}_{\text{2D-track}}$. In **Step 5**, we use OMR [73] for $\mathcal{F}_{\text{lane}}$. In **Step 6**, Metric3D [74] is used for $\mathcal{F}_{\text{depth}}$ and SAM [75] for $\mathcal{F}_{\text{seg}}$. In **Step 7**, we again use InternVL for $\mathcal{F}_{\text{I-VLM}}$. In **Step 8**, we use CenterTrack [76] for $\mathcal{F}_{\text{3D-det}}$. For peer-informed reasoning, VideoLLaMA2 [14] serves as the default peer model unless otherwise specified. For final reasoning step, we use GPT-4o-mini [77] for $\mathcal{F}_{\text{LLM}}$. The full list of hyperparameters and prompts is in the Supplementary Material. *Note that* our goal is not to compare model variants, but to demonstrate the effectiveness of the combined vision–language pipeline. Each can be easily swapped for stronger versions for further improvements.

**Experiment details.** We choose datasets and baselines that require all the methods to analyze the complete video before answering the user query. To this end, we use four benchmarks: MM-AU (Multi-Modal Accident Video Understanding) [22], SUTD (Traffic Question Answering) [78], LingoQA [79], and Nexar [80] dataset. For baselines, we compare *iFinder* with V-VLMs that are trained to provide open-ended responses to users' queries on input videos. In particular, we compare

Table 1: **Multiple-choice VQA performance on MMAU and SUTD.** *pt* = prompt tuning. *i***Finder** beats both all V-VLMs w/o fine-tuning, showing that fine-grained details generate more accurate choices.

| Method | MM-AU | SUTD |
|---|---|---|
| Generalist Models | | |
| VideoLLaMA2 [14] | 50.95 | 47.51 |
| VideoLLaMA2 (w/ pt) | 52.89 | - |
| VideoChat2 [21] | 49.56 | 42.17 |
| Video-LLaVA [15] | 43.63 | 38.35 |
| Driving-specialized Methods | | |
| DriveMM [16] | 24.22 | 43.90 |
| *i***Finder** (Ours) | **63.39** | **50.93** |

Table 2: **Result on SUTD categories.** Basic Understanding (U), Event Forecasting (F), Reverse Reasoning (R), Counterfactual Inference (C), Introspection (I), and Attribution (A).

| Method | U | F | R | C | I | A |
|---|---|---|---|---|---|---|
| Generalist Models | | | | | | |
| VideoLLaMA2 [14] | 49.2 | 39.0 | 48.5 | 53.5 | 35.8 | 45.2 |
| VideoChat2 [21] | 42.5 | 38.1 | 43.8 | 49.2 | 30.4 | 42.8 |
| Video-LLaVA [15] | 39.7 | 37.2 | 35.8 | 40.5 | 31.1 | 36.4 |
| Driving-specialized Methods | | | | | | |
| DriveMM [16] | 47.6 | 38.6 | 40.1 | 43.2 | 38.5 | 37.7 |
| *i***Finder** (Ours) | **52.2** | **43.5** | **50.2** | **56.8** | **39.2** | **49.6** |

Table 3: **Open-ended VQA result on LingoQA dataset.** *i***Finder** outperforms others on the Lingo-Judge accuracy without fine-tuning.

| Method | Lingo-J | BLEU | METEOR | CIDEr |
|---|---|---|---|---|
| Generalist Models | | | | |
| VideoLLaMA2 [14] | 36.00 | 4.15 | 33.45 | 26.28 |
| VideoChat2 [21] | 41.20 | **6.58** | **36.81** | 40.98 |
| Video-LLaVA [15] | 21.00 | 4.26 | 26.99 | 31.23 |
| Driving-specialized Methods | | | | |
| WiseAD [18] | 13.40 | 2.20 | - | 21.50 |
| *i***Finder** (Ours) | **44.20** | 6.07 | 35.80 | **42.01** |

Table 4: **Accident Occurrence Prediction on Nexar dataset.** VideoLLaMA2 and WiseAD exhibit a bias toward predicting accidents in all cases. We use VideoChat2 in peer-informed reasoning, further enhancing its performance.

| Method | Acc (%) | F1-Score | Precision | Recall |
|---|---|---|---|---|
| Generalist Models | | | | |
| VideoChat2 [21] | 58.0 | 0.62 | 0.57 | 0.68 |
| VideoLLaMA2 [14] | 50.0 | **0.67** | 0.50 | **1.00** |
| Driving-specialized Methods | | | | |
| DriveMM [16] | 49.0 | 0.32 | 0.48 | 0.24 |
| WiseAD [18] | 50.0 | **0.67** | 0.50 | **1.00** |
| *i***Finder** (Ours) | **62.0** | 0.59 | **0.64** | 0.54 |

against recent state-of-the-art general V-VLMs VideoLLaMA2 [14], VideoChat2 [21], and Video-LLaVA [15], as well as those proposed for driving video analysis DriveMM [16] and WiseAD [18]. The dataset and evaluation metric details are provided in the Supplementary Material. All the following evaluations are in a strictly zero-shot setting, with no fine-tuning.

**Quantitative Results.** We analyze the performance on accident-cause and traffic scene understanding on MM-AU and SUTD datasets under multi-choice VQA setup, shown in Table 1 and Table 2, and gain three insights on *i***Finder** versatility and performance. *One*, *i***Finder** outperforms both general-purpose and driving-specialized models on MM-AU and SUTD without any fine-tuning, highlighting its strong out-of-the-box reasoning capabilities. *Two*, it achieves the best accuracy in all six SUTD categories, from basic understanding to attribution, showing robust generalization across diverse cognitive tasks. *Three*, the 39-point gap between *i***Finder** and DriveMM on MM-AU underscores the limitations of current domain-specific models and the strength of *i***Finder**'s architecture in complex, real-world driving scenarios.

Next, we analyzed all methods under an open-ended VQA setup on the LingoQA dataset in Table 3 and provided the following insights. *One*, *i***Finder** achieves the best Lingo-Judge accuracy among all evaluated methods, outperforming VideoLLaMA2 by 3%, VideoChat2 by 8%, and Video-LLaVA by 23.2%. Further, it outperforms driving-specialized models, WiseAD by 30.8%. *Two*, while our method performs competitively on BLEU and METEOR metrics, it slightly lags behind VideoChat2. This suggests that VideoChat2 generates responses that are more lexically similar to reference answers but do not necessarily reflect greater factual correctness. We also show this in Figure 5.

Finally, we analyze all methods for accident occurrence prediction on the Nexar dataset in Table 4. *i***Finder** achieves the highest accuracy at 62.0%, outperforming both generalist models, as well as driving-specialized models. Notably, while VideoLLaMA2 and WiseAD achieve high F1 and perfect recall by *always* predicting accidents, this limits real-world reliability. In contrast, VideoChat2 balances precision and recall for better accuracy. Building on this, *i***Finder** integrates it within the peer V-VLM to enhance both accuracy and precision.

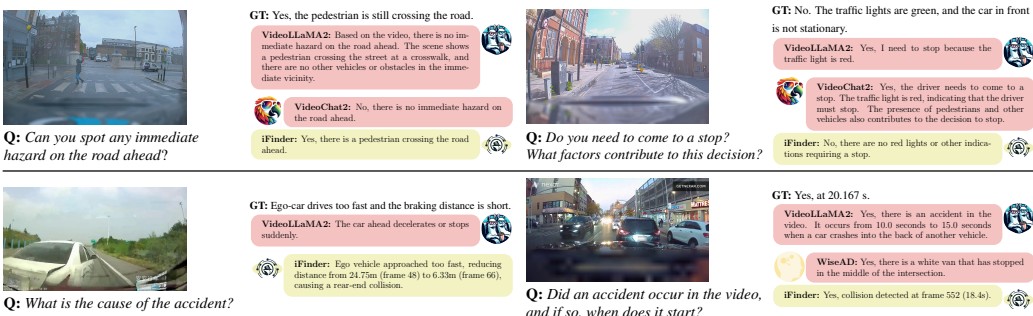

Figure 5: **Qualitative comparison on LingoQA (*top*), MM-AU (*bottom*, *left*), and Nexar (*bottom*, *right*) dataset. *i*Finder** improves spatial reasoning and causal inference, and reduces peer-V-VLM errors. In the bottom-left example, *i*Finder corrects the peer V-VLM's inaccurate claim of a "decelerated vehicle" by leveraging structured data that reveals the ego vehicle's rapid approach.

Table 5: **Impact of each vision component and each prompt block on MM-AU dataset.**

| | Method | Accuracy (%) |
|---|---|---|
| | VideoLLaMA2 (peer-VLM) | 52.89 |
| vision | *i*Finder w/o ego-state estimation | 62.37 |
| | *i*Finder w/o lane detection | 61.80 |
| | *i*Finder w/o distance estimation | 60.62 |
| | *i*Finder w/o frame undistortion | 60.47 |
| | *i*Finder w/o object attributes | 59.04 |
| | *i*Finder w/o orientation estimation | 58.83 |
| | *i*Finder w/o scene understanding | 57.81 |
| prompt | *i*Finder w/o peer instruction | 60.62 |
| | *i*Finder w/o steps instruction | 60.06 |
| | *i*Finder w/o key explanation | 58.73 |
| | *i*Finder | **63.39** |

Table 6: **Result breakdown on MMAU weather categories. *i*Finder** achieves best performance across adversarial conditions, showing its adaptability to diverse environments.

| Method | Foggy | Rainy | Snowy | Sunny |
|---|---|---|---|---|
| Generalist Models | | | | |
| VideoLLaMA2 (w/pt) [14] | 66.67 | 44.83 | 40.25 | 54.56 |
| VideoChat2 [21] | 58.33 | 54.31 | 49.69 | 49.16 |
| Video-LLaVA [15] | 58.33 | 41.38 | 42.77 | 43.76 |
| Driving-specialized Methods | | | | |
| DriveMM [16] | 33.33 | 24.14 | 20.13 | 24.55 |
| *i*Finder (Ours) | **75.00** | **65.52** | **57.86** | **63.69** |

**Qualitative Results.** We qualitatively analyze all methods in Figure 5 and observe the following. *One*, *i*Finder demonstrates superior perceptual grounding—for instance, in the first case, unlike VideoLLaMA2 and VideoChat2, *i*Finder correctly identifies the pedestrian as an immediate hazard, aligning with the ground truth. *Two*, *i*Finder shows fine-grained causal reasoning: in the second row, *i*Finder specifies that the ego vehicle reduced its distance from 24.75m to 6.33m, clearly linking this to the collision—something missing in the generic response by VideoLLaMA2. *Three*, *i*Finder exhibits higher temporal precision, as seen in the final example where *i*Finder accurately detects the collision at frame 552 (18.4s), closely matching the GT (20.167s), while others offer vague or less aligned time windows. These examples underscore *i*Finder's robust visual grounding and decision-making fidelity.

**Ablation studies.** In Table 5, we assess the impact of each vision component, as well as the contribution of each prompt component in $P_{LLM}$ on the MM-AU dataset. Although each vision component contributes to overall performance, surprisingly, scene understanding and orientation estimation are of higher importance for semantic reasoning than even core perception modules like distance or lane estimation. For example, removing orientation estimation leads to the largest drop in accuracy—from 63.39% to 58.83%. The results show that each block aid accuracy, with *key explanations* being especially crucial—without them, the LLM sometimes generates invalid responses, such as mis-formatted numerical outputs.

**System analysis under adversarial conditions.** Post-hoc driving video analysis also encompasses scenarios involving adversarial or long-tail conditions, such as poor weather or nighttime environments. To evaluate the robustness of our method under such conditions, we break down the performance across different weather and lighting conditions using the MM-AU[22] benchmark in Table 6 and Table 7. While generalist models (*e.g.*, VideoLLaMA2[14]) show steep drops in performance under rain (44.83%), snow (40.25%), and night (50.23%), and driving-specialized models like DriveMM perform poorly (*e.g.*, 20.13% on snowy, 24.14% on rainy), *i*Finder maintains consistently high accuracy—achieving the best scores across all settings.

Table 7: **Result breakdown on MMAU light categories.** *i***Finder** shows consistent improvements across different lighting conditions, indicating its robustness to variations in illumination.

| Method | Day | Night |
|---|---|---|
| Generalist Models | | |
| VideoLLaMA2 (w/pt)[14] | 53.23 | 50.23 |
| VideoChat2[21] | 50.29 | 43.84 |
| Video-LLaVA[15] | 43.25 | 46.58 |
| AD-specialized Methods | | |
| DriveMM[16] | 23.41 | 30.59 |
| *i***Finder** (Ours) | **63.32** | **63.93** |

Table 8: **Error propagation analysis of** *i***Finder on MM-AU.** We evaluate *i***Finder** under increasing confidence thresholds $\tau_{score}$ to simulate missed detections, showing strong resilience to error propagation.

| Method | Accuracy (%) | Objects Retained(%) |
|---|---|---|
| *i***Finder** | **63.39** | 100.00 |
| *i***Finder** w $\tau_{score} = 0.4$ | 63.13 | 82.48 |
| *i***Finder** w $\tau_{score} = 0.5$ | 61.09 | 32.63 |
| *i***Finder** w $\tau_{score} = 0.6$ | 58.73 | 7.96 |
| *i***Finder** w $\tau_{score} = 0.7$ | 58.42 | 3.16 |
| *i***Finder** w $\tau_{score} = 0.8$ | 58.27 | 0.68 |

**Impact of error propagation by object detection.** With the object detector being the earliest and most important module to capture object semantics in *i***Finder**, we analyze the impact of error propagation or missed detections occurring early in the system on the overall performance in Table 8. It can be observed that even when fewer than 1% of objects are retained (0.68% at confidence score $\tau_{score} = 0.8$), the accuracy remains at 58.27%, better than the best baseline (VideoLLaMA2[14] at 52.89%). At $\tau_{score} = 0.5$, retaining only 32.63% of objects still yields 61.09% accuracy, suggesting that *i***Finder**'s reasoning remains stable under significant perceptual filtering. Overall, *i***Finder** demonstrates graceful degradation and limited error propagation.

# 5 Conclusion

In this paper, we argue that grounding general-purpose LLMs using structured, interpretable scene representations offers a more accurate alternative than V-VLMs. Further, decoupling perception from reasoning enables a more reliable analysis system. Built on these principles, we introduce *i***Finder**, a modular, training-free framework that advances post-hoc driving video analysis by structurally grounding general-purpose LLMs in domain-specific perception. Through explicit scene decomposition and hierarchical prompting, *i***Finder** achieves superior spatial and causal reasoning—outperforming both generalist and driving specialized V-VLMs on accident reasoning benchmarks. Extensive evaluations confirm strong post-hoc video understanding performance under diverse environmental conditions. This structured grounding method highlights a promising direction for aligning LLMs with domain-specific reasoning requirements, where interpretability and reliability constitute the primary focus.

# 6 Acknowledgments and Disclosure of Funding

This work was supported by NSF grant CNS-2312395 and NEC Laboratories America, Inc.

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

Table 9: Licenses and sources of datasets and models used in this work.

| Asset | License |
|---|---|
| MM-AU[22] | CC-BY-NC-4.0 |
| SUTD-TrafficQA [78] | Custom (research-only, non-commercial) |
| LingoQA [79] | https://github.com/wayveai/LingoQA (allowed for research) |
| Nexar [80] | nexar-open-data-license |
| VideoLLaMA2 [14] | Apache-2.0 |
| VideoChat2 [21] | MIT |
| VideoLLaVA [15] | Apache |
| DriveMM [16] | Apache-2.0 |
| WiseAD [18] | MIT |
| GPT-4o-mini [77] | This code uses GPT-4o-mini via OpenAI's API. The model itself is proprietary and subject to OpenAI's Terms of Use. |

- If assets are released, the license, copyright information, and terms of use in the package should be provided. For popular datasets, paperswithcode.com/datasets has curated licenses for some datasets. Their licensing guide can help determine the license of a dataset.
- For existing datasets that are re-packaged, both the original license and the license of the derived asset (if it has changed) should be provided.
- If this information is not available online, the authors are encouraged to reach out to the asset's creators.

13. **New assets**

Question: Are new assets introduced in the paper well documented and is the documentation provided alongside the assets?

Answer: [NA]

Justification: [NA]

Guidelines:

- The answer NA means that the paper does not release new assets.
- Researchers should communicate the details of the dataset/code/model as part of their submissions via structured templates. This includes details about training, license, limitations, etc.
- The paper should discuss whether and how consent was obtained from people whose asset is used.
- At submission time, remember to anonymize your assets (if applicable). You can either create an anonymized URL or include an anonymized zip file.

14. **Crowdsourcing and research with human subjects**

Question: For crowdsourcing experiments and research with human subjects, does the paper include the full text of instructions given to participants and screenshots, if applicable, as well as details about compensation (if any)?

Answer: [NA]

Justification: [NA]

Guidelines:

- The answer NA means that the paper does not involve crowdsourcing nor research with human subjects.
- Including this information in the supplemental material is fine, but if the main contribution of the paper involves human subjects, then as much detail as possible should be included in the main paper.
- According to the NeurIPS Code of Ethics, workers involved in data collection, curation, or other labor should be paid at least the minimum wage in the country of the data collector.

15. **Institutional review board (IRB) approvals or equivalent for research with human subjects**

    Question: Does the paper describe potential risks incurred by study participants, whether such risks were disclosed to the subjects, and whether Institutional Review Board (IRB) approvals (or an equivalent approval/review based on the requirements of your country or institution) were obtained?

    Answer: [NA]

    Justification: [NA]

    Guidelines:

    - The answer NA means that the paper does not involve crowdsourcing nor research with human subjects.
    - Depending on the country in which research is conducted, IRB approval (or equivalent) may be required for any human subjects research. If you obtained IRB approval, you should clearly state this in the paper.
    - We recognize that the procedures for this may vary significantly between institutions and locations, and we expect authors to adhere to the NeurIPS Code of Ethics and the guidelines for their institution.
    - For initial submissions, do not include any information that would break anonymity (if applicable), such as the institution conducting the review.

16. **Declaration of LLM usage**

    Question: Does the paper describe the usage of LLMs if it is an important, original, or non-standard component of the core methods in this research? Note that if the LLM is used only for writing, editing, or formatting purposes and does not impact the core methodology, scientific rigorousness, or originality of the research, declaration is not required.

    Answer: [Yes]

    Justification: We disclose comprehensive information about the language model, including its official name and the specific prompts utilized.

    Guidelines:

    - The answer NA means that the core method development in this research does not involve LLMs as any important, original, or non-standard components.
    - Please refer to our LLM policy (https://neurips.cc/Conferences/2025/LLM) for what should or should not be described.

Table 10: Wall-clock runtime of each pipeline module in *i*Finder.

| Module | Runtime(s) |
|---|---|
| Frame Undistortion | 66.7 |
| 3D Object Detection | 14.8 |
| Attribute Estimation | 66.7 |
| Distance Estimation | 40.3 |
| Lane Detection | 21.5 |

## A  Limitations & Broader Impacts

**Limitations.** The current form of *i*Finder lacks mechanisms to incorporate or reason about ambiguous, social, or normative aspects of driving scenes (*e.g.*, intent behind a maneuver, yielding behavior, *etc*.). These elements are often critical in understanding traffic interactions but not easily captured by purely spatial-temporal features or symbolic grounding. Future work should build this capability using hybrid reasoning mechanisms that combine structured perceptual data with commonsense knowledge bases.

**Broader Impacts.** On the positive side, by clearly separating how the system sees the world (perception) from how it thinks about it (reasoning), *i*Finder supports a growing trend in AI that large language models (LLMs) have limits and need structured, trustworthy data to reason well. This makes the system's thinking more like human reasoning, which is especially important in high-stakes areas like self-driving cars. On the negative side, although *i*Finder improves clarity and explainability, it might unintentionally promote a narrow view of knowledge where only LLM-readable information is seen as valid. This could leave out important human factors that are harder to define, like a driver's intent, ethical responsibility, or social rules of the road.

## B  Implementation choices

In **Step 3**, we sample the temporal points in order to reduce noise and make the estimation insensitive to small deviations. Further, we set $\tau_a$ and $\tau_s$ as 30° and standard deviation of all speeds $\{s_t\}_{t=0}^{T}$. For motion estimation, we set $g$ as 2. In **Step 4**, since we use Owl-V2, we set the 2D classes as ['motorcycle', 'police car', 'ambulance', 'bicycle', 'traffic light', 'stop sign', 'road sign', 'construction worker', 'police officer', 'ambulance', 'fire truck', 'construction vehicle', 'traffic cone', 'person', 'car', 'wheelchair', 'bus', 'truck'] with confidence threshold as 0.25. In **Step 5**, we only estimate lane locations for vehicles and person categories. In **Step 8**, we use the default classes by Center-Track [76] for NuScenes dataset [81]. All the rest of the parameters are set as default model choices. All the prompts are provided in Section E. Note that for peer V-VLM, we use the default prompt provided by the respective authors. All experiments were conducted on a single NVIDIA A6000 GPU with 48 GB of memory. Table 10 reports the average wall-clock time required to execute each module in our pipeline per video on the LingoQA dataset [79]. These timings reflect end-to-end processing, including loading, inference, and output serialization (for *unoptimized* python code).

## C  Dataset Details

**Dataset Details.** MMAU contains a test set of 1,953 ego-view accident videos, each associated with a fixed question: *"What is the cause of the accident?"* along with five multiple-choice answer options. The SUTD-TrafficQA dataset includes a test set of 4,111 real-world driving videos, paired with 6,075 multiple-choice questions designed to assess different aspects of scene understanding. The questions are categorized into six reasoning types: Basic Understanding, which involves direct perception of scene elements; Event Forecasting, which requires predicting future events; Reverse Reasoning, which focuses on deducing past events from the current scene; Counterfactual Inference, which evaluates hypothetical scenarios; Introspection, which involves providing preventive advice; and Attribution, which involve causal reasoning and responsibility assessment in driving scenarios. Performance on both datasets is measured using accuracy. For open-ended VQA, we evaluate on LingoQA [79], which consists of 100 videos with a total of 500 questions in the evaluation set.

Table 11: List of sampled videos from the Nexar dataset

```
01031, 00831, 00097, 02034, 01080, 01085, 01736, 00059, 02121, 01875,
01970, 01290, 00967, 01840, 00477, 01853, 00469, 00970, 01815, 02085,
00684, 00587, 01393, 02013, 00816, 01858, 01607, 00534, 02048, 00407,
01806, 01586, 00077, 01413, 00099, 01478, 00858, 00155, 01801, 01276,
02119, 01350, 01696, 00364, 01616, 01753, 00039, 01682, 00783, 01992,
01932, 01372, 01638, 01268, 01542, 00049, 01617, 00904, 02069, 00640,
00046, 00106, 00937, 01465, 00579, 00131, 01118, 00703, 00324, 00339,
00167, 01635, 00103, 01695, 00608, 00949, 00422, 01317, 00610, 00242,
00519, 00909, 01952, 01364, 01071, 00461, 01453, 01849, 01533, 00345,
00733, 00617, 00722, 00453, 01985, 00651, 00972, 01441, 00977, 00082
```

Table 12: Standard error of *i***Finder** on MM-AU and SUTD datasets.

| Method | MM-AU | SUTD |
|---|---|---|
| *i***Finder** (ours) | 63.39±0.26 | 50.93±0.68 |

Unlike MM-AU [22] and SUTD-TrafficQA [78], which follow a multiple-choice format, LingoQA [79] requires free-form natural language responses. For accident occurrence prediction, we evaluate on the Nexar dataset [80]. Since the original test set does not include ground-truth labels, we randomly sample 100 videos from the training set to construct an evaluation set, maintaining a balanced distribution of 50 accident and 50 non-accident videos, consistent with the ratio in the full training set. The list of sampled videos from the Nexar dataset's original training set, used for accident occurrence prediction evaluation in this paper, is provided in Table 11. The list indicates the video names.

# D   Error Analysis

In Table 12, performance of *i***Finder** on MM-AU and SUTD datasets, reported as (mean accuracy ± standard error) over five runs. We can observe that *i***Finder** maintains its performance.

Although *i***Finder** achieves strong results, it struggles in visually ambiguous collisions. For example, from the Nexar dataset, video "01031", a car cuts in front of the ego vehicle, but without clear cues such as steering correction or vibration, even humans cannot confirm contact; this reveals *i***Finder** 's reliance on observable patterns rather than imperceptible dynamics. In the video "00970", the ego car brakes sharply behind another vehicle, leaving only a small gap. With no visible impact signs like deformation or debris, it is unclear whether this was a collision or a near-miss. Both cases show that subtle physical contact often leaves no visual trace, limiting vision-only approaches. Future work should integrate other modalities (e.g., audio, IMU) and model uncertainty so the system can express lower confidence instead of making forced binary predictions.

# E   Prompts

The prompts we use for the VLM are shown in Listing 1, Listing 2 and Listing 3. The system and user prompts we use for each of the tasks in the final reasoning are shown in Listing 4, Listing 5, Listing 6, Listing 7, and Listing 8.

Listing 1: Prompt for image-based VLM $P_I$ in **Step 2**.

```
You are an expert in autonomous driving, specializing in analyzing traffic scenes.
    You receive a series of traffic images from the perspective of the ego car.
    Your task is to describe the driving environment, focusing on weather, lighting
    , road layout, surrounding environment, and any notable elements.

It is essential that you strictly follow the rules and instructions below. Any
    deviation from the specified structure or format will result in an invalid
    output.
```

```
STRICTLY follow Rules:
 - You must strictly follow the dictionary structure provided below.
 - Only use the specified terms for weather, light, road layout, and environment. Do
      not create your own terms.
 - No additional information or categories should be added.
 - You should strictly follow these instructions. If an object or element is not
      visible or does not exist in the scene, set the value to 'None'. Ensure every
      field is filled with the appropriate value or 'None'.
 - For the video description, base your analysis on the overall characteristics
      observed throughout the video rather than a single frame.
 - Note any temporal changes that occur over time in the video (e.g., traffic flow
      shifts, traffic light changes, road condition variations).

Output the result in the following dictionary format:

{
  "surrounding_info": {
    "weather": "[e.g., 'cloudy', 'sunny', 'rainy', 'fog', 'snowy']",
    "light": "[Choose 'day', 'night', 'dawn', 'dusk']",
    "road_layout": "[Choose from: 'straight road', 'curved road', 'intersection', 'T
        -junction', 'ramp']",
    "environment": "[Choose from: 'city street', 'country road', 'highway', '
        residential area']",
    "sun_visibility_conditions": "[Choose from: 'clear', 'foggy', 'low visibility',
        'hazy']",
    "road_condition": "[Choose from: 'wet', 'icy', 'normal', 'debris', 'potholes']",
    "surface_type": "[Choose from: 'asphalt', 'gravel', 'dirt', 'concrete']",
    "traffic_flow": "[Choose from: 'light', 'moderate', 'heavy']",
    "time_of_day": "[Choose from: 'morning', 'afternoon', 'evening', 'night']",
    "road_obstacles": "[Choose from "debris visible", "no debris visible".]"
    "road_density": "[Choose from "crowded", "normal", 'scarce'.]"
    "
    },
  "description": "[Provide a concise yet informative summary of the scene. Highlight
        notable objects, traffic conditions, movement patterns, mention any
      observable changes over time, and any potential driving hazards.]"
}
```

Listing 2: Prompt for video-based VLM $P_V$ in **Step 2**.

```
Analyze the provided driving video and generate a detailed, sequential caption that
    accurately describes the vehicle's actions, road conditions, traffic dynamics,
    and surrounding environmental elements. Highlight key driving events, such as
    acceleration, braking, turning, interactions with other vehicles or pedestrians
    , and the presence of traffic signals, signs, or notable landmarks.
    Additionally, provide an in-depth analysis of the ego car's speed, discussing
    its impact on the scene and how it influences the behavior and dynamics of
    nearby objects and road users.
```

Listing 3: Prompt for video-based VLM $P_d$ in **Step 7**.

```
You are an expert in autonomous driving, specializing in analyzing traffic scenes.
    You are driving the ego-vehicle and looking at the scene.

Your task is to look at the red bounding box and output the response in the format
    below. If it is a person, say "person wearing black clothes", etc. If it is any
    other vehicle, say "black car", "black bus", "silver SUV", etc.
Strictly follow the rules.

{
    "color": "[Choose the most dominant color of this object.]"
}
```

Listing 4: System prompt in $P_{\text{LLM}}$ for multiple-choice VQA.

```
You are a detailed traffic analyst, analyzing scene data to draw fact-based
    conclusions about vehicle behavior, lane positions, and potential hazards.
```

Listing 5: User prompt in $P_{\text{LLM}}$ for multiple-choice VQA.

```
You are analyzing a JSON data file representing a traffic scene from the ego vehicle
    's perspective. The video captures interactions with surrounding objects.
    Analyze only observable elements: bounding boxes ('bbox'), lane positions ('
    relative_lane_location', 'obj_lane_location', 'ego_lane_location'), object
    rotations ('rot_y'), distances ('distance_from_ego_vehicle'), attributes ('
    attributes'), and additional environmental factors from "Video Level
    Information" such as weather, lighting, and road conditions.
---
JSON Key Explanations
- "bbox": Represents the detected object's position in the frame. Track changes in
    size and location to determine motion and distance.
- "distance_from_ego_vehicle": Distance (in meters) from the ego vehicle.
- "relative_lane_location": Description of how many lanes away an object is from the
    ego vehicle.
- "obj_lane_location": Object's lane index relative to the road.
- "ego_lane_location": Ego vehicle's lane index relative to the road.
- "attributes": Object features such as color.
- "rot_y": Object's rotation angle, useful for detecting turns.
- "loc": Object's position in 3D space.
- "object_id": Unique identifier for objects in each frame.
- "surrounding_info": Describes the environment, including weather, lighting, road
    layout, surface type, traffic flow, and time of day.
- "motion_state": Indicates the motion status (e.g., Moving, Stopped) of the ego
    vehicle.
- "turn_action": Describes the ego vehicle's turning behavior.
- "description": Summary of the video.
- "response": Response from another model; it may be incorrect, but use it as a
    basis for reasoning.
---
Instructions:
Follow the steps below to analyze the incident and formulate your response using
    JSON data and the description under "Video Level Information" to enhance
    reasoning. Think step by step and use the exact format specified at the end.
---
Step 1: Identify and Describe the Unusual Activity or Event
Step 1.1: Analyze the following data points to identify risky or dangerous behaviors
    :
- Bounding box ('bbox'): Track object movements and changes in size or proximity.
- Lane position ('obj_lane_location'): Detect lane changes or encroachments.
- Rotation ('rot_y'): Identify unusual rotation patterns suggesting erratic or risky
    behavior.
 - Distance: Measure the proximity of objects to the ego vehicle.

Describe any patterns or anomalies, such as:
- Objects moving against traffic.
- Lane cutting or abrupt merging.
- Unusual or sudden changes in distance or rotation.
- Sharp or erratic rotations ('rot_y'), e.g., sharp spinning of a vehicle indicating
    slipping on an icy or wet road.

Use specific data points to explain behaviors:
- If a vehicle shows sharp changes in rotation ('rot_y') on icy or wet roads,
    classify it as "Vehicle slipping off-road due to wet/icy conditions."
- If a vehicle moves across lanes unexpectedly into the ego vehicle's path, classify
    it as "Lane cutting or forceful merging incident."
- If an object (e.g., a pedestrian, animal, or vehicle) suddenly enters the ego
    vehicle's path at close proximity, classify it accordingly (e.g., "Unexpected
    pedestrian crossing in front of ego vehicle").
```

```
Step 1.2:
Based on your analysis, classify the incident using the following examples:
1. Lane cutting or forceful merging incident.
2. Close-proximity vehicle or pedestrian crossing in front of the ego vehicle.
3. Vehicle collision.
4. Vehicle slipping off-road due to wet/icy conditions.
5. Traffic rule violation encounter.
6. Unexpected animal crossing in front of the ego vehicle.
---
Step 2: Provide Potential Reason for the Incident

Identify the possible reason why the incident occurred. The reason must be based on
    specific observable data in the JSON file. Use information such as:
- Lane changes.
- Rotation angles ('rot_y').
- Object proximity to the ego vehicle.
- Environmental indicators like weather conditions.
- Other "video-level" information if included.

Step 3: Choose the Best Explanation from the Given Options
Based on the JSON data, select the most appropriate option that best explains the
    cause of the incident.
---
Key Requirements for Your Response:
1. Select only one of the given multiple-choice options as the final answer.
3. Do not generate an open-ended response. The final answer must be exactly one
    option from the provided list.
4. Base your answer strictly on observable data (bounding boxes, lane positions,
    rotations, distances, etc.).
5. If the data is incomplete, make an educated guess but still select the most
    appropriate option.
6. Never state "not enough information" or "unable to determine". You must always
    pick the most reasonable answer.
7. Format your final answer exactly as specified-just the letter corresponding to
    your choice.

Answer the question precisely and analytically based only on the observable data in
    the provided JSON file.
---
Response Format (Strictly Follow This Format):
    [Letter]
---
JSON Data:
{JSON data}
---
{Question}
Options: {Options}
Answer with the option's letter from the given choices directly and only give the
    best option. The best answer is:
```

Listing 6: System prompt in $P_{\text{LLM}}$ for open-ended VQA and accident occurrence prediction.

```
You are a detailed traffic analyst, analyzing scene data to draw fact-based
    conclusions about vehicle behavior, lane positions, and potential hazards.
    Focus on elements such as bounding boxes ('bbox'), lane changes, rotation ('
    rot_y'), and proximity to the ego vehicle. Use these attributes to form precise
     insights, noting any deviations from normal behavior, changes in object
    orientation, or risky maneuvers.
```

Listing 7: User prompt in $P_{\text{LLM}}$ for open-ended VQA.

```
You are analyzing a JSON file representing a traffic video from the ego vehicle's
    perspective. The video captures interactions with surrounding objects. Analyze
    only observable elements: bounding boxes ('bbox'), lane positions ('
    relative_lane_location', 'obj_lane_location', 'ego_lane_location'), object
```

        rotations ('rot_y'), distances ('distance_from_ego_vehicle'), attributes ('
        attributes'), and additional environmental factors from "Video Level
        Information" such as weather, lighting, and road conditions.
Prioritize later frames for analysis.
---
Key Rules
- Focus on later frames for all interpretations.
- Use common knowledge where applicable:
1. Traffic lights can only show one color at a time.
2. An object very far (e.g., 50+ meters) from the ego car is not considered in the
      ego lane.
- For color-related questions:
1. Check the latest frames first.
2. If multiple colors exist, return only the most frequent color from later frames.
3. Return exactly ONE color. Never list multiple colors.
---
JSON Key Explanations
- "bbox": Represents the detected object's position in the frame. Track changes in
      size and location to determine motion and distance.
- "distance_from_ego_vehicle": Distance (in meters) from the ego vehicle.
- "relative_lane_location": Description of how many lanes away an object is from the
       ego vehicle.
- "obj_lane_location": Object's lane index relative to the road.
- "ego_lane_location": Ego vehicle's lane index relative to the road.
- "attributes": Object features such as color.
- "rot_y": Object's rotation angle, useful for detecting turns.
- "loc": Object's position in 3D space.
- "object_id": Unique identifier for objects in each frame.
- "surrounding_info": Describes the environment, including weather, lighting, road
      layout, surface type, traffic flow, and time of day.
- "motion_state": Indicates the motion status (e.g., Moving, Stopped) of the ego
      vehicle.
- "turn_action": Describes the ego vehicle's turning behavior.
- "description": Summary of the video.
- "respone": Response from another model; it may be incorrect, but use it as a basis
       for reasoning.
---
Step-by-Step Analysis
Step 1: Identify Key Event
- Analyze movements using later frames first.
- Categorize the event as lane change, pedestrian crossing, cyclist movement,
      turning vehicle, steady lane position, traffic sign, unexpected object, or
      other notable behavior.

Step 2: Provide One Reason (10 Words)
- Provide one reason, exactly 10 words, using JSON data and the description under "
      Video Level Information" to enhance reasoning.

Step 3: Answer as a Driver
- Do NOT mention JSON metadata (IDs, raw values).
- Answer naturally like a driver.
- Yes/No questions: Give a direct, brief explanation.
- Fact-based questions: Base response on visible elements.
- Color-related questions: Return only ONE dominant color from later frames.
---
Key Constraints
1. Analyze later frames first.
2. Use common knowledge (traffic light rules, far objects not in ego lane).
3. For color: Pick ONE most frequent color from later frames.
4. NEVER list multiple colors. Always return a single color.
5. Make no assumptions beyond the data.
---
JSON Data:
{JSON data}
---

```
{Question}
---
```

Listing 8: User prompt in $P_{\mathrm{LLM}}$ for accident occurrence prediction.

```
You are analyzing a JSON file representing a traffic video from the ego vehicle's
    perspective. The video captures interactions with surrounding objects. Analyze
    only observable elements: bounding boxes ('bbox'), lane positions ('
    relative_lane_location', 'obj_lane_location', 'ego_lane_location'), object
    rotations ('rot_y'), distances ('distance_from_ego_vehicle'), attributes ('
    attributes'), and additional environmental factors from "Video Level
    Information" such as weather, lighting, and road conditions. Your goal is to
    determine whether an accident occurs and, if so, identify the frame index where
     it begins.
---
JSON Key Explanations
- "bbox": Represents the detected object's position in the frame. Track changes in
     size and location to determine motion and distance.
- "distance_from_ego_vehicle": Distance (in meters) from the ego vehicle.
- "relative_lane_location": Description of how many lanes away an object is from the
     ego vehicle.
- "obj_lane_location": Object's lane index relative to the road.
- "ego_lane_location": Ego vehicle's lane index relative to the road.
- "attributes": Object features such as color.
- "rot_y": Object's rotation angle, useful for detecting turns.
- "loc": Object's position in 3D space.
- "object_id": Unique identifier for objects in each frame.
- "surrounding_info": Describes the environment, including weather, lighting, road
    layout, surface type, traffic flow, and time of day.
- "motion_state": Indicates the motion status (e.g., Moving, Stopped) of the ego
    vehicle.
- "turn_action": Describes the ego vehicle's turning behavior.
- "description": Summary of the video.
- "respone": Response from another model; it may be incorrect, but use it as a basis
     for reasoning.
---
Step-by-Step Analysis
Step 1: Identify Key Event
- Categorize the event as lane change, pedestrian crossing, cyclist movement,
    turning vehicle, steady lane position, traffic sign, unexpected object, or
    other notable behavior.
- Check "motion_state" for abrupt stops and "bbox" overlaps for potential collisions
    .

Step 2: Determine If an Accident Occurs
- Always return within 10 words.
- Start with "Yes" or "No".
- If an accident is detected, provide:
   1. The frame number or timestamp of occurrence.
   2. Example: "Yes, collision detected at frame 600."
- If no accident occurred:
   1. Example: "No."
- If the data is incomplete, make an educated guess.
   1. Never state "not enough information" or "unable to determine"-you must always
        pick one from "Yes" or "No".
---
Key Constraints
1. Do NOT mention raw numerical data from the JSON, except for the frame index.
2. Make no assumptions beyond the data.
---
JSON Data:
{JSON data}
---
Did an accident occur in the video, and if so, when does it start (provide a frame
    index)?
```

---

## F   Example of Extracted Data Structure

Figure 6 presents an example of data extracted from the corresponding video (LingoQA dataset) using *i***Finder**.

## G   Additional Qualitative Results

We provide additional qualitative comparisons in Figure 7 and Figure 8 to further illustrate the advantages of *i***Finder** over baseline methods. Figure 7 shows two examples where *i***Finder** corrects the peer V-VLM. Figure 8 shows two examples where *i***Finder** shows better grounded responses to users' questions compared to baselines.

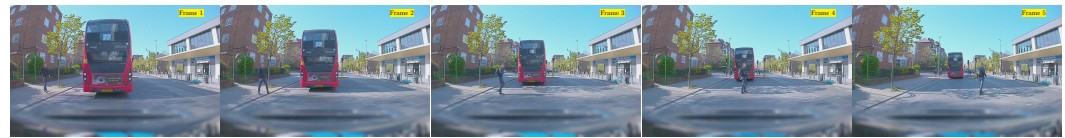

```
{
    "Video Level Information": {
        "surrounding_info": {
            "weather": "sunny",
            "light": "day",
            "road_layout": "straight road",
            "environment": "city street",
            "sun_visibility_conditions": "clear",
            "road_condition": "normal",
            "surface_type": "asphalt",
            "traffic_flow": "light",
            "time_of_day": "morning",
            "road_obstacles": "no debris visible",
            "road_density": "normal"
        },
        "ego-car-information": {
            "frame_index: 0": {
                "motion_state": "Moving",
                "turn_action": "Straight"
            },
            "frame_index: 1": {
                "motion_state": "Stopped",
                "turn_action": "Straight"
            },
            "frame_index: 2": {
                "motion_state": "Stopped",
                "turn_action": "Straight"
            },
            "frame_index: 3": {
                "motion_state": "Stopped",
                "turn_action": "Straight"
            },
            "frame_index: 4": {
                "motion_state": "Stopped",
                "turn_action": "Straight"
            }
        },
        "description": "The video depicts a sunny day with clear visibility on a
            city street. The road is straight and devoid of any debris. The traffic
            flow is light, with a red double-decker bus and a pedestrian crossing
            the road. The surrounding environment includes buildings, trees, and a
            bus stop. The scene is typical of a morning commute with no significant
            hazards or obstructions.",
        "response": "The current action is a car driving down a street. The
            justification is that the car is moving forward on the road."
    },
    "Frame Level Information": [
        {
            "frame_index": 0,
```

continued from previous page ...

```
        "detected_objects": [
            {
                "class": "bus",
                "bbox": [677,106,1229, 875], "object_id": 1,
                "distance_from_ego_vehicle": "7.41 meters",
                "relative_lane_location": "same lane as ego vehicle",
                "attributes": "Red color",
                "tracking_id": 4
            },
            {
                "class": "person",
                "bbox": [180,541,333, 871], "object_id": 11,
                "distance_from_ego_vehicle": "7.70 meters",
                "relative_lane_location": "same lane as ego vehicle",
                "attributes": "wearing Black clothes",
                "loc": [-4.62, 1.33, 8.54], "rot_y": -1.64,
                "tracking_id": 1
            },
            {
                "class": "bicycle",
                "bbox": [1516,650,1540, 686], "object_id": 9,
                "distance_from_ego_vehicle": "41.14 meters"
            },
            {
                "class": "bicycle",
                "bbox": [1567,670,1616, 731], "object_id": 10,
                "distance_from_ego_vehicle": "22.75 meters"
            }],
    {
        "frame_index": 1,
        "detected_objects": [
            {
                "class": "bus",
                "bbox": [811,218,1087, 816], "object_id": 2,
                "distance_from_ego_vehicle": "8.72 meters",
                "relative_lane_location": "same lane as ego vehicle",
                "attributes": "Red color",
                "tracking_id": 4
            },
            {
                "class": "bicycle",
                "bbox": [1567,670,1616, 731], "object_id": 10,
                "distance_from_ego_vehicle": "27.22 meters"
            },
            {
                "class": "person",
                "bbox": [180,541,333, 871], "object_id": 11,
                "distance_from_ego_vehicle": "12.62 meters",
                "relative_lane_location": "same lane as ego vehicle",
                "attributes": "wearing Black clothes",
                "loc": [-3.8, 1.31, 8.0], "rot_y": -1.0,
                "tracking_id": 1
            },
        ]
    },
```

continued from previous page ...

```
        {
            "frame_index": 2,
            "detected_objects": [
                {
                    "class": "bus",
                    "bbox": [902,335,1107,758],
                    "object_id": 3,
                    "distance_from_ego_vehicle": "12.01 meters",
                    "relative_lane_location": "same lane as ego vehicle",
                    "attributes": "Red color",
                    "loc": [0.23, 1.26, 19.88], "rot_y": -1.43,
                    "tracking_id": 4
                },
                {
                    "class": "traffic light",
                    "bbox": [1245,581,1258, 608], "object_id": 5,
                    "distance_from_ego_vehicle": "79.30 meters",
                    "attributes": "Green light"
                },
                {
                    "class": "bicycle",
                    "bbox": [1512,647,1534, 678], "object_id": 9,
                    "distance_from_ego_vehicle": "12.37 meters"
                }
            ]
        },
        {
            "frame_index": 3,
            "detected_objects": [
                {
                    "class": "bus",
                    "bbox": [900,405,1067, 734], "object_id": 2,
                    "distance_from_ego_vehicle": "16.71 meters",
                    "relative_lane_location": "same lane as ego vehicle",
                    "attributes": "Red color",
                    "rot_y": -1.52, "loc": [-0.36, 1.46, 24.36],
                    "tracking_id": 4
                },
                {
                    "class": "person",
                    "bbox": [959,561,1048, 862], "object_id": 3,
                    "distance_from_ego_vehicle": "8.41 meters",
                    "relative_lane_location": "same lane as ego vehicle",
                    "attributes": "wearing Black clothes",
                    "loc": [-0.16, 1.47, 10.16], "rot_y": -0.31,
                    "tracking_id": 1
                },
                {
                    "class": "traffic light",
                    "bbox": [814,514,834, 570], "object_id": 19,
                    "distance_from_ego_vehicle": "27.23 meters",
                    "attributes": "Green light"
                },
            ]
        },
```

```json
    {
        "frame_index": 4,
        "detected_objects": [
            {
                "class": "bus",
                "bbox": [925,458,1056, 713], "object_id": 2,
                "distance_from_ego_vehicle": "22.00 meters",
                "relative_lane_location": "same lane as ego vehicle",
                "attributes": "Red color",
                "loc": [-0.14, 1.46, 32.59], "rot_y": -1.53,
                "tracking_id": 4
            },
            {
                "class": "traffic light",
                "bbox": [805,518,828, 569], "object_id": 19,
                "distance_from_ego_vehicle": "26.02 meters",
                "attributes": "Green light"
            },
            {
                "class": "person",
                "bbox": [959,680,1013, 872], "object_id": 3,
                "distance_from_ego_vehicle": "10.78 meters",
                "relative_lane_location": "same lane as ego vehicle",
                "attributes": "wearing Black clothes"
            },
        ]
    }]
}
```

Figure 6: *i*Finder JSON data structure for LingoQA dataset video shown above

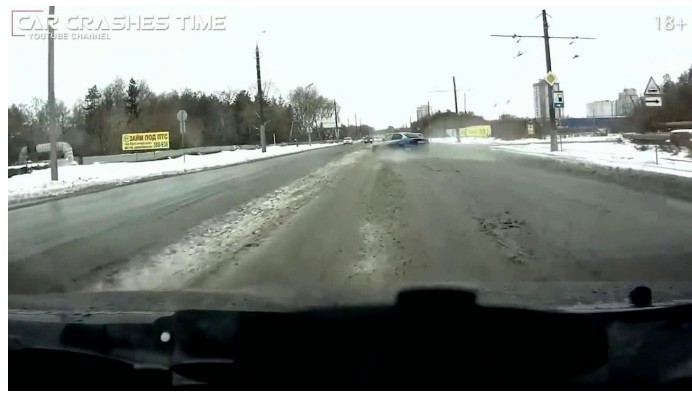

**Q:** *What is the cause of the accident??*

**GT:** The car does not notice the coming vehicles when crossing the road.

> **VideoLLaMA2:** The car ahead decelerates or stops suddenly. 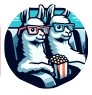

> **iFinder:** Car veered into ego-vehicle's lane, causing potential collision incident. 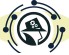

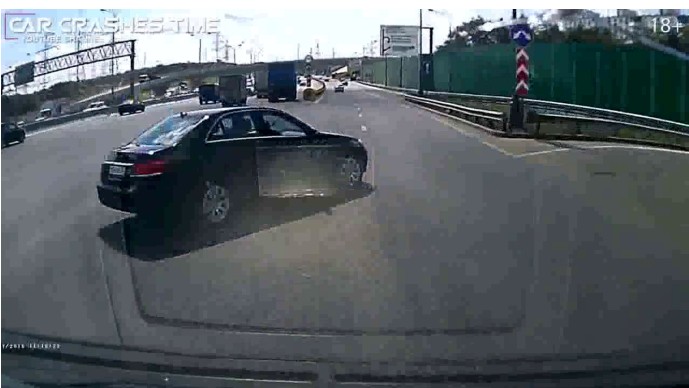

**Q:** *What is the cause of the accident?*

**GT:** The car does not notice the coming vehicles when crossing the road.

> **VideoLLaMA2:** The car's vision is blocked and there is no time to brake. 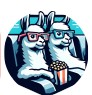

> **iFinder:** Black car (ID:6) abruptly merges into ego vehicle's lane causing a near-collision. 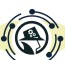

Figure 7: Qualitative visualization where *i*Finder corrects peer V-VLM on MMAU dataset.

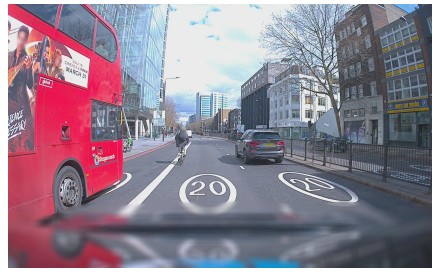

**Q:** *Is there traffic in your lane? If so, what types of vehicles are present??*

**GT:** Yes, a cyclist.

**VideoLLaMA2:** Yes, there is traffic in the lane. The video shows a red double-decker bus and other vehicles such as cars and trucks on the road. 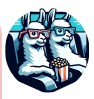

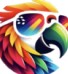 **VideoChat2:** Yes, there is traffic in the lane. The vehicles present include a red double-decker bus, a black car, and a blue van.

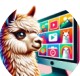 **VideoLLaVA:** Yes, there is traffic in the lane. The video shows cars and a bus driving down the street.

**iFinder:** Yes, there is a cyclist. 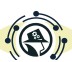

---

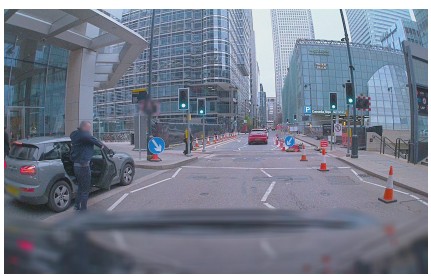

**Q:** *Is there a traffic light in sight? If so, what color is it exhibiting?*

**GT:** Yes, green.

**VideoLLaMA2:** Yes, there is a traffic light visible in the video. It appears to be displaying a red light. 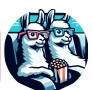

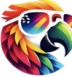 **VideoChat2:** Yes, there is a traffic light in sight. It is exhibiting a red light.

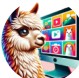 **VideoLLaVA:** Yes, there is a traffic light in sight, and it is displaying a red light.

**iFinder:** Yes, there is a traffic light in sight, and it is exhibiting a green light. 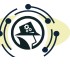

Figure 8: Qualitative comparison against baselines on LingoQA dataset.

