# OpenReview forum: "iFinder: Structured Zero-Shot Vision-Based LLM Grounding for Dash-Cam Video Reasoning"
_NeurIPS.cc/2025/Conference — NeurIPS 2025 poster_

### Official Review · Reviewer_qGHL · 2025-06-02

**Clarity:** 3
**Significance:** 2
**Originality:** 2
**Rating:** 4
**Confidence:** 4

**Summary:**

The paper introduces a structured semantic grounding framework, iFinder, to improve zero-shot vision-based reasoning of dash-cam driving videos by utilizing the reasoning capabilities of large language models. Instead of depending on end-to-end vision-language models (that are capable of video processing), this framework separates perception from reasoning. It uses pretrained vision models to extract driving-related information, such as object detection, lane position, and depth estimation. The framework then organizes the driving-related information and feeds this information to an LLM via structured prompts. This approach enables step-wise, grounded reasoning, leading to more accurate and interpretable driving scenario analyses than conventional VLMs, particularly in accident causation and hazard detection.

**Questions:**

Q1. How do you determine the prompts? I.e., how do you know the current prompts are "optimal"? Are there any ablation studies with varied prompts?

Q2. How does the choice of the LLM (in Fig. 2) affect the performance?

Q3. Can this framework be generalized to other domains? Or, will the framework still work if I remove a subset of modules (e.g., remove 2D object detection + tracking and keep the remaining modules)? What if I add more modules (e.g., global latitude & longitude)?

Q4. A follow-up for Q3: Do you need to refine the prompts or retrain the LLM when adding or removing modules?

Q5: Is this framework adaptable to multi-view cameras? E.g., self-driving cars have six cameras at different angles.

**Ethical Concerns:**

["NO or VERY MINOR ethics concerns only"]

**Final Justification:**

The authors have addressed my concerns and added the necessary results. So, I decide to increase my score.

**Limitations:**

L1. The authors mention that real-time deployment is not a concern of this work. However, the authors could have a discussion on the applications of this framework beyond real-time deployment. I'm not clear where this framework can be applied except for real-time deployment.

L2. While the paper includes some analysis on error propagation (showing graceful degradation when object detection confidence is low), it would be valuable to see a deeper discussion about how sensitive the entire pipeline is to failures or inaccuracies in upstream modules (e.g., tracking, depth estimation). How does ı̇Finder handle compounding errors, and are there mechanisms for uncertainty quantification or correction? Is there a way to provide probabilistic guarantees to the framework's outcomes?

L3. The framework's performance heavily relies on the accuracy of multiple off-the-shelf vision models (e.g., for detection, tracking, lane estimation, depth, etc.). The overall reasoning can degrade if these modules perform poorly due to unusual conditions or domain shift.

L4. The paper lacks a discussion of limitations and future work in the main body.

**Paper Formatting Concerns:**

There are no formatting concerns.

**Quality:**

3

**Strengths And Weaknesses:**

**Strengths**
- The paper provides structured, symbolic cues (such as object orientation, lane context, and object distance) to the LLM, enabling transparent reasoning, making the analysis process more interpretable and trustworthy for complex driving scenarios.
- Sufficient empirical analysis shows that the framework significantly outperforms both general-purpose and driving-specific VLMs on multiple zero-shot driving benchmarks.
- The framework uses a modular design based on pretrained vision models, which allows easy integration and parallelization of new perception modules without retraining.

**Weaknesses**
- The framework heavily relies on the quality of the existing deep learning models.
- The framework cannot provide any guarantees for their outcomes.

*Note to the authors*: More details of the weaknesses are in **Questions** and **Limitations**, so please respond to those sections ONLY during the rebuttal.

---

> ### Author Rebuttal · Authors · 2025-07-31
>
> Thank you for the insightful review of our paper. Below are our responses to your questions. Please do not hesitate to raise further questions.
>
> ----
>
> **Responses to Questions** \
> **Response to Q1.** We agree that prompt design plays a crucial role in LLM-based reasoning and clarify the following:
>  - Prompts are designed based on functional decomposition. As detailed in Sec. 3.1.2, our three-block prompt (Key Explanation, Step Instructions, Peer Response) is motivated by established findings in structured prompting and step-wise LLM reasoning [60–63]. It explicitly aligns with the hierarchical scene representation, guiding the LLM to reason over symbolic inputs.
> -  Yes, we perform prompt ablations in Table 6 and provide quantitative ablations over prompt components. Removing "Key Explanation" drops MM-AU accuracy by 4.66% (63.39 → 58.73), "Step Instructions" drops performance by 3.33%, and "Peer Instruction" drops performance by 2.77%.
> - Prompt design was not searched for optimality. Rather, it is guided by alignment between input structure and reasoning goals, and validated empirically via ablations (in Table 6). For example, if we drop a module from iFinder for ablation, we removed it's corresponding explanation from the main prompt
>
>
> ----
> **Response to Q2.** Based on the reviewer's recommendation, we analyze the LLM choice on iFinder and present the analysis below (MM-AU dataset).
> | LLM                    | Acc (%) |
> |------------------------|---------|
> | GPT-4o-mini (in paper) | 63.39   |
> | GPT-4.1-mini           | 63.65   |
> | GPT-4.1-nano           | 52.53   |
>
>
>
> ----
> **Response to Q3.**
>  - **Generalization to other domains:** Yes, the iFinder framework can be generalized to other domains. This is because the iFinder pipeline is explicitly designed as a modular, training-free framework to facilitate flexibility in incorporating or removing specific vision components. The core idea remains the same: representational abstraction of vision information that enables symbolic reasoning over dynamic scenes in language via LLM. The schema—temporal object-centric data—is domain-structured, not domain-locked. Its modular design allows substitution of perception modules for other domains (e.g., indoor robotics), and the JSON format can accommodate arbitrary scene cues. Our ablations (Table 6) show performance is robust to component removal, indicating generalizable reasoning.
>  - **Adding /Removing modules to iFinder:** As shown in our method (Section 3.1) and ablation results (Table 6), each module contributes uniquely to the final reasoning output, and modules can be swapped or extended without retraining. As aforementioned,   adding or removing modules would only affect the instructions sent to the LLM to understand the JSON file.
>
>
>
> ----
> **Response to Q4.** We only need to change the prompt to include/remove information about added/removed modules for the LLM to understand the JSON file. No retraining of LLM is needed.
>
> ----
> **Response to Q5.** Yes, ı̇Finder can be adapted to multi-view camera setups. Since our framework is modular and operates on per-frame visual inputs, each view can be processed independently through the same perception modules. The resulting structured cues (e.g., object pose, orientation, depth) from each camera can then be merged into a unified symbolic representation using view-specific metadata (e.g., extrinsics). While our current work focuses on monocular front-view videos, extending the pipeline to multi-view settings involves modifying the scene representation to accommodate cross-view spatial alignment—an area we identify as promising future work for enhancing coverage and reducing occlusion in complex scenarios.
>
>
> ----
> ----
> **Responses to Limitations** \
> **Response to L1.** We thank the reviewer for raising this point. While we explicitly mention that real-time deployment is not a focus of our work, the primary goal of our framework is to enable post-hoc, structured analysis of driving scenes—a task that is fundamentally distinct from real-time inference. Our method is particularly suited for **offline** applications such as accident investigation, insurance assessment, driver behavior auditing, and fleet-level safety analytics, where *interpretability, causal attribution*, and *verifiable reasoning* are critical. As shown in our evaluations on MM-AU, SUTD, and Nexar datasets, our framework excels in post-event reasoning tasks, outperforming both generalist and driving-specific VLMs in accuracy and robustness. For example, On the MM-AU dataset—focused explicitly on accident reasoning—iFinder achieves 63.39% accuracy, significantly outperforming the best generalist V-VLM (VideoLLaMA2 with prompt tuning) at 52.89%, and the best driving-specialized model (DriveMM) at 24.22%.
>
> ----
> **Response to L2.**
>
> - We analyzed the impact of error propagation (or effect of compounding errors when the initial module, in our case the object detector, fails to detect objects in the scene) in Table 7 on MM-AU dataset. For example, to imitate this detection failure, we remove >97% of object detections (τ = 0.8) from the JSON files and observed that the overall accuracy still remains 58.27%, surpassing all baselines including VideoLLaMA2 (52.89%) (Table 7). This shows that our symbolic reasoning retains reliability under perceptual degradation. Further, Table 7 also shows that iFinder demonstrates graceful degradation and limited error propagation. Please see L326-333 for a detailed discussion.
>  - Also based on Reviewer zz3N ‘s suggestion, we performed a different error propagation analysis: replacing modules with their weaker counterparts for the object tracker and the depth estimation on MM-AU dataset. We observed that replacing the tracker with a weaker tracker drops performance slightly to 61.70%, indicating that tracking quality contributes moderately to system accuracy. Similarly, using a weaker depth model causes a minor drop to 62.78%.
>  - Kindly note that while uncertainty quantification is not explicitly implemented in the current version of iFinder, the underlying LLM can support it. For instance, methods like Monte Carlo dropout, and token-level entropy estimation in the LLM’s outputs could be integrated to estimate uncertainty. This is a promising direction for future work. Further, discussing probabilistic guarantees for iFinder’s outcomes is currently beyond the scope of the paper.
>
>
> ----
> **Response to L3.** While iFinder integrates multiple vision modules, its robustness to degraded perception is quantitatively demonstrated. As shown in Table 5, iFinder outperforms all baselines under adverse conditions—achieving 65.52% in rain, 57.86% in snow, and 63.93% at night, while the best generalist model (VideoLLaMA2) drops to 44.83%, 40.25%, and 50.23%, respectively. These results underscore that iFinder’s structured grounding mitigates failure cascades and ensures stable, domain-shift-tolerant reasoning.
>
>
> ----
> **Response to L4.** Thank you for highlighting this. Owing to space constraints, we included these discussions in the supplementary material. We will incorporate them into the main body of the manuscript in the final version.
>
> ----

---

> > ### Author Response · Authors · 2025-08-04
> >
> > Hi Reviewer qGHL,
> >
> > Thank you again for reviewing our work and insightful comments. We wanted to check if our responses have clarified your questions. Your support is extremely valuable to us.
> >
> > Best wishes, \
> > Authors

---

> > ### Comment · Reviewer_qGHL · 2025-08-05
> >
> > Thanks for providing additional results and explanations. The rebuttal has addressed the majority of my concerns.
> >
> > I have some follow-up questions: As the framework uses LLM as an intermediate reasoning step, what is the latency? How will the latency change due to the video length or multi-camera view?
> >
> > The experiments are focusing on traffic scenarios. Is this framework able to operate in real time?
> >
> > The questions above may be beyond the scope of this work. I will increase my score to 4 regardless of the answer to the questions above. However, I think it would be beneficial to include the latency statistics in the paper and discuss how to reduce them.

---

> > > ### Author Response · Authors · 2025-08-06
> > >
> > > Hi Reviewer qGHL,
> > >
> > > Thank you for taking the time to read our responses. We are glad that the responses addressed majority of your concerns. Here are the responses to your follow-up questions:
> > >
> > > ----
> > >  - **Latency from LLM:** For the LingoQA dataset, which contains shorter videos, the LLM takes an average of 0.63 seconds per query. As video length increases—as in the MMAU dataset—the response time is an average of 0.74 seconds.
> > >
> > > - **Is this framework able to operate in real time?:** We can convert iFinder to real-time depending on compute available as follows-
> > >     - **Limited Compute (e.g., few GPUs):** Replace each model with a more efficient alternative to reduce latency.
> > >     - **Sufficient Compute (e.g., 8×A6000 GPUs):** Host API versions of the models on a GPU node. The iFinder vision pipeline can then query these APIs and retrieve JSON outputs. Additionally, enable parallel execution of independent modules, and aggregate their outputs into a final JSON.
> > >
> > > ----
> > > We **sincerely thank** the reviewer for their consideration and revising the score to 4. We greatly value the feedback provided and will incorporate all comments in the revised version of the paper.

---

### Official Review · Reviewer_XWwS · 2025-06-03

**Clarity:** 3
**Significance:** 2
**Originality:** 2
**Rating:** 4
**Confidence:** 4

**Summary:**

ı̇Finder is a zero-shot framework that grounds LLMs for driving video analysis by decoupling perception and reasoning. It converts dash-cam videos into structured, interpretable representations, capturing object motion, lane position, and orientation, etc. It also uses them to guide LLMs in accurate, causal reasoning, outperforming baselines on multiple benchmarks.

**Questions:**

1. What is the contributions of this paper?
2. What is the inferencing time and memory usage during inference?
3. What is the potential direction to improve the efficiency based on the current design?
4. Why driving-specialized baselines performance pretty bad?

**Ethical Concerns:**

["NO or VERY MINOR ethics concerns only"]

**Final Justification:**

I acknowledge the contribution of this work in enabling symbolic reasoning over dynamic scenes using LLMs, which is an interesting and promising direction. The incorporation of more efficient sub-modules is also a practical design choice that helps alleviate the efficiency concerns to some extent.

That said, the technical novelty of the paper appears somewhat limited. The proposed approach primarily integrates existing components in a relatively straightforward manner, which diminishes the strength of its technical contribution.

In summary, I will raise my score to a 4, as most of my concerns have been addressed. However, I still believe that the technical depth of the paper is limited, despite the overall direction being compelling.

**Limitations:**

I think the largest limitation of the method is the inference latency, which is not mentioned in the limitation in the supplementary materials.

**Paper Formatting Concerns:**

No.

**Quality:**

2

**Strengths And Weaknesses:**

Strengths:
1. This paper is well-written and easy to follow.
2. This method requires no training and has good interpretability.
3. This method achieves competitive results among several benchmarks.

Weakness
1. My main concern is that the paper’s contribution is not clearly defined or significant. There is no explicit contribution statement, and the introduction only outlines three experimental findings without a clear methodological or conceptual innovation. This makes it difficult to assess the novelty and impact of the work.
2. This paper introduces seven additional intermediate processing steps for dash-cam video reasoning, but nearly all of them rely on separate off-the-shelf models to extract specific cues. While the pipeline is training-free, the use of multiple models makes the system quite heavy. I am curious about the overall memory usage during inference, as excessive memory demands could make the approach impractical for real-time or resource-constrained driving scenarios.
3. From Lines 252 to 255, the paper states that real-time performance is not the focus, emphasizing interpretability and accuracy instead. However, in driving scenarios, inference efficiency is just as critical as accuracy. The proposed method relies on seven intermediate modules and a peer V-VLM for final LLM processing, which raises concerns about latency and scalability. If the improved accuracy comes at the cost of significantly slower inference, the practical value of the approach becomes questionable. At least, the paper should report and compare inference time with existing baselines. Additionally, it would strengthen the work to propose a possible direction for improving efficiency based on the current design.
4. In Tables 1–3, the driving-specialized baselines (DriveMM and WiseAD) perform significantly worse than generalist models. Notably, DriveMM achieves only 49% accuracy in Table 3, which is worse than random guessing (50%). This raises concerns about whether these methods are truly strong baselines or representative of the state-of-the-art driving-specialized  methods in driving video understanding.

---

> ### Author Rebuttal · Authors · 2025-07-31
>
> Thank you for the thoughtful and encouraging review of our paper. Below are our responses to your questions. Please do not hesitate to raise further questions.
>
> ----
>
> **Response to Weakness 1 & Question 1.** The conceptual contribution of ı̇Finder is the idea of a representational abstraction of vision inputs that enables symbolic reasoning over dynamic scenes using LLMs. In other words, rather than passing raw pixel inputs or latent embeddings to the LLM, we transform each dash-cam video into a structured, temporally indexed JSON format. This representation bridges the gap between low-level perception and high-level reasoning, acting as a semantic interface between vision modules and language models.
>
> ----
> **Response to Weakness 2 & Question 2.** We agree that ı̇Finder is not optimized for real-time deployment. As stated in the end of Sec. 3.1.2, we emphasize that ı̇Finder targets interpretability-critical scenarios, not embedded real-time inference. To clarify:
>  - Note that modular design ≠ high memory usage. All modules run sequentially or in parallel without maintaining persistent GPU state. Peak memory remains below 48 GB on an A6000 GPU (see Appendix B). The largest modules—frame undistortion and object attribute extraction—are image-based and not memory-bound.
>  - Wall-clock runtime is acceptable for post-hoc use. The full pipeline processes a LingoQA video in <4.5 minutes with unoptimized Python (Table 1 in supplementary material), suitable for offline analysis.
>  - We can achieve scalability via decoupling. Each module can be independently swapped for faster or lower-memory alternatives (e.g., lighter depth or 3D estimators), offering scalability based on deployment needs.
>
> We will clarify this use-case focus in the final version.
>
>
> ----
> **Response to Weakness 3 & Question 3.**
>  - As iFinder is built on top of correcting peer V-VLM (VLMs designed for high-level reasoning), it incurs higher computational cost relatively. However, this cost is justified by a notable increase in performance, particularly in tasks requiring nuanced understanding and disambiguation. For example in LingoQA dataset (Table 2), VideoLLama2 and VideoLLaVA incur 2.07 sec and 2.04 sec, respectively. However, iFinder (with 165 sec) outperforms them by 8.2% and 23.2%, respectively. Similarly under adversarial conditions (Table 5) in "Foggy", "Rainy", "Snowy" and "Night" conditions, iFinder outperforms them by (8.33%, 20.69%, 17.61%, 13.7%) and (16.67%, 24.14%, 15.09%, 17.35%), respectively. The improved accuracy and robustness validate the extremely significant value of leveraging vision-language grounding offered by iFinder.
>
>  - There are two approaches to improve the framework’s efficiency:
>     - **Limited Compute (e.g., few GPUs):** Replace each model with a more efficient alternative to reduce latency.
>     - **Sufficient Compute (e.g., 8×A6000 GPUs):** Host API versions of the models on a GPU node. The iFinder vision pipeline can then query these APIs and retrieve JSON outputs. Additionally, enable parallel execution of independent modules, and aggregate their outputs into a final JSON.
>
>
> ----
> **Response to Weakness 4 & Question 4.**  We understand the concern and clarify the following:
>  - DriveMM [16] and WiseAD [18] are SOTA domain-specialized baselines for ego-centric driving scene understanding. We use the official implementations and pre-trained checkpoints. DriveMM (49%) and WiseAD (50%) on Nexar (Table 3) do not perform worse than chance; the class distribution for the experiment is 50/50 by design (see Appendix C in supplementary material). We investigated this performance further and observed high recall + low precision (e.g., 1.00 recall, 0.50 precision in WiseAD) which indicates bias towards predicting accidents (reported in Table 3).
>  - Further note that, generalist models outperform driving-specialized models due to massive video pre-training. Models like VideoChat2 and VideoLLaMA2 benefit from scaling, not task specificity. That they outperform DriveMM highlights a gap in domain-adapted VLMs, a key motivation for ı̇Finder’s structured grounding.
>
> ----

---

> > ### Author Response · Authors · 2025-08-04
> >
> > Hi Reviewer XWwS,
> >
> > Thank you again for reviewing our work and encouraging words. We wanted to check if our responses have clarified your questions. Your support is extremely valuable to us.
> >
> > Best wishes, \
> > Authors

---

> > ### Comment · Reviewer_XWwS · 2025-08-05
> > **Reply for rebuttal**
> >
> > I acknowledge the contribution of this work in enabling symbolic reasoning over dynamic scenes using LLMs, which is an interesting and promising direction. The incorporation of more efficient sub-modules is also a practical design choice that helps alleviate the efficiency concerns to some extent.
> >
> > That said, the technical novelty of the paper appears somewhat limited. The proposed approach primarily integrates existing components in a relatively straightforward manner, which diminishes the strength of its technical contribution.
> >
> > In summary, I will raise my score to a 4, as most of my concerns have been addressed. However, I still believe that the technical depth of the paper is limited, despite the overall direction being compelling.

---

> > > ### Author Response · Authors · 2025-08-05
> > >
> > > Hi Reviewer XWwS,
> > >
> > > We sincerely thank the reviewer for their constructive feedback and for acknowledging the significance of our work in enabling symbolic reasoning over dynamic scenes using LLMs. We also appreciate the recognition of our suggested design choices aimed at improving efficiency. We raise the following points to the reviewer's closing comments:
> > >
> > > ----
> > > - Regarding the concern about technical novelty, we respect the reviewer’s perspective. Our goal was to demonstrate how introducing a representational abstraction of visual inputs—effectively decoupling low-level perception from high-level reasoning—can enable symbolic reasoning over dynamic scenes, thereby allowing LLMs to be grounded in domain-specific tasks, such as interpreting driving videos.
> > > - While the components (e.g., object detection, depth estimation, 3D pose) are pretrained, integrating them into a coherent, temporally grounded, LLM-compatible structured data format requires non-trivial design choices:
> > >     - The system aligns outputs across different modalities and resolutions (2D, 3D, semantic, temporal) into a hierarchical JSON schema that maintains per-frame and global consistency—something standard perception stacks don’t support.
> > >     - Many raw outputs (like camera pose, depth, rot_y) are converted into interpretable abstractions (e.g., “left turn,” “cutting lanes,” “object in ego lane”), requiring careful rule design and spatial reasoning (e.g., Eq. 4–7, 10–11 in Section 3.1.1).
> > >
> > >     So, while the components are off-the-shelf, the design, alignment, abstraction, and reasoning orchestration are technically sophisticated and far from trivial.
> > >
> > > ----
> > > We are **extremely grateful** for the reviewer’s thoughtful evaluation and for raising the score to 4 in light of our rebuttal. We will incorporate all comments in the revised version of the paper.
> > >
> > > Best wishes, \
> > > Authors

---

### Official Review · Reviewer_XuKK · 2025-07-03

**Clarity:** 4
**Significance:** 4
**Originality:** 3
**Rating:** 6
**Confidence:** 3

**Summary:**

The paper presents iFinder, a framework for zero-shot post-hoc reasoning on dash-cam videos. It decouples perception and reasoning by converting raw video into structured, hierarchical data (including object orientation, lane position, and ego motion), which is then passed to a general-purpose LLM. Such an approach enables interpretable, grounded reasoning and significantly improves performance over existing V-VLMs on 4 benchmarks. Despite its strong results, the possible use of the proposed framework is most likely limited "just" to offline analysis.

**Questions:**

* _Q1:_ Could the authors provide rough estimates of pipeline inference time and resource requirements? Even _coarse_ timing would be useful.
* _Q2:_ Is there a plan to extend ı̇Finder toward semi-real-time scenarios (e.g., batch processing at the edge)? If not, what would be the major technical problem?
* _Q3:_ How adaptable is the pipeline to other domains beyond driving? Could the same framework be applied to indoor navigation or surveillance video with minor changes?
* _Q4:_ Would the authors consider evaluating ı̇Finder on more challenging or domain-shifted settings such as those proposed in [1]? These benchmarks focus on out-of-distribution hazard identification and symbolic reasoning, and would align well with ı̇Finder’s strengths.
* _Q5:_ Since many LLMs are inherently non-deterministic, could the authors clarify whether multiple runs of the reasoning stage produce consistent outputs? Given the structured symbolic input and multi-step prompt design, some variability is expected, but it would be helpful to know if the system’s answers are stable in practice. Did the authors use any measures (e.g., fixed temperature, response filtering, or prompt anchoring) to manage output variability?

**Ethical Concerns:**

["NO or VERY MINOR ethics concerns only"]

**Final Justification:**

The authors have addressed my main concerns with clear responses and additional results. If included in the camera-ready, it will largely improve the quality and transparency of the paper.

Reading the other reviewers’ comments did not change my mind about the contributions of the provided paper, and I will keep my rating.

**Limitations:**

Mostly yes. The authors acknowledge the offline nature of iFinder and briefly mention its lack of real-time deployment support. However, several additional limitations deserve discussion.
1. The system relies on many pre-trained models, each of which may introduce noise, yet, no formal analysis of this sensitivity is provided.
2. The scene representation is handcrafted, which limits the method’s generalization.
3. While the system claims interpretability and robustness, no formal evaluation or consistency analysis is conducted to support it.
4. The results are most likely based on single-seed evaluation without statistical significance testing

**Paper Formatting Concerns:**

I have not noticed any major formatting issues.

**Quality:**

4

**Strengths And Weaknesses:**

**Note**: _Despite I'm a little bit "negative" in the review, I believe this is a strong and well-written paper. The design, structured reasoning approach, and strong zero-shot results make it an extremely valuable contribution. While there are areas for improvement, I view these more as opportunities for future work than fundamental flaws._

### **Strengths**
- **STRUCTURE and LANGUAGE**: The paper follows a standard structure, which makes it easier to follow and read. Overall, the paper is well-written. The level of English is great. Figures are of good quality and appropriately support the clarity. Formally, everything is on a high level.

- **REPRODUCIBILITY**: Although the code is not provided, the paper and supplementary materials present outstanding implementation details, making it potentially "easy" to reproduce. Despite the large complexity and inherently non-deterministic nature of some LLMs, I believe there is no better way how to do it.

- **METHOD DESIGN**: The core idea is well-designed and implemented. The framework is modular, does not require additional fine-tuning/training, and introduces a principled way to integrate domain knowledge into LLM reasoning via hierarchical scene representations and well-thought-out prompting.

- **STRONG PERFORMANCE**: The method achieves consistent and significant improvements across four benchmarks, outperforming both general-purpose and domain-specific video-language models in zero-shot settings. Notably, the "performance" gains are much higher in tasks requiring spatial and causal reasoning.

- **RELEVANCE**: The paper addresses a highly relevant problem in vision-language understanding for not just autonomous systems. Post-hoc reasoning from dash-cam videos is a challenging yet increasingly important task, especially given the limitations of real-time data collection in real-world driving environments.

### **Weaknesses**

- **SYSTEM COMPLEXITY**: The framework relies on many pretrained models (e.g., for object detection, depth estimation, 3D orientation, lane detection, ego-motion), which increases complexity and potential points of failure. Besides, the paper does not provide any system-level runtime or memory analysis, making it more challenging to assess practical deployment costs.

- **RELIANCE ON RULE-BASED STRUCTURE**: The scene representation is handcrafted and domain-specific. While effective for driving, it’s unclear how easily this structure transfers to other domains (e.g., indoor navigation, robotics), or how it would perform under noisy or ambiguous visual inputs.

- **UNCLEAR PEER CONTRIBUTION**: The peer V-VLM is treated as a noisy first-pass guess, but its actual contribution to the final performance is not deeply analyzed. A deeper ablation (e.g., replacing the peer) would clarify whether it meaningfully supports the reasoning or serves as a distractor.


**[Additional Comments]:**
- The Related Work section is rather short. While key methods are covered,  recent works on structured reasoning and OOD evaluation [1, 2, 3], particularly those addressing zero-shot hazard identification under domain shift, are missing, while it could be beneficial to include them. Evaluating iFinder on such a dataset/benchmark would significantly strengthen the paper’s narrative around robustness and generalization. Given the method’s zero-shot, modular nature, this would be a natural and valuable extension.
- The qualitative examples are useful and illustrate the reasoning capabilities well. However, it would be valuable to also include some failure cases with a brief analysis, especially where the LLM misinterprets the structured input or the visual modules produce conflicting cues.
- The authors apparently misunderstand what is expected regarding statistical significance. Reporting results from a single seed is insufficient, even if followed by a qualitative error analysis. Statistical significance typically requires multiple runs with different random seeds and corresponding variance or confidence intervals. The current justification ("we provide an error analysis in the Supplementary Material") does not address it. Although I understand the computation cost is rather heavy, a few repeated runs with 2–3 random seeds would make the results more trustworthy and support stronger conclusions.

**[Grammar & Typos & Formating]**:
- Line 240: The JSON example is overflowing the line width.
- Table 1-7: The random verticall allignement does not look apealing. While minor, it feels like a huge ink stain on an otherwise good-looking and polished submission.

___________________________________

[1] @article{alshami2024coool,
  title={Coool: Challenge of out-of-label a novel benchmark for autonomous driving},
  author={AlShami, Ali K and Kalita, Ananya and Rabinowitz, Ryan and Lam, Khang and Bezbarua, Rishabh and Boult, Terrance and Kalita, Jugal},
  journal={arXiv preprint arXiv:2412.05462},
  year={2024}
}

[2] @inproceedings{picek2025zero,
  title={Zero-shot hazard identification in Autonomous Driving: A Case Study on the COOOL Benchmark},
  author={Picek, Lukas and Cermak, Vojtech and Hanzl, Marek},
  booktitle={Proceedings of the Winter Conference on Applications of Computer Vision},
  pages={654--663},
  year={2025}
}

[3] @article{shriram2025towards,
  title={Towards a multi-agent vision-language system for zero-shot novel hazardous object detection for autonomous driving safety},
  author={Shriram, Shashank and Perisetla, Srinivasa and Keskar, Aryan and Krishnaswamy, Harsha and Bossen, Tonko Emil Westerhof and M{\o}gelmose, Andreas and Greer, Ross},
  journal={arXiv preprint arXiv:2504.13399},
  year={2025}
}

---

> ### Author Rebuttal · Authors · 2025-07-31
>
> Thank you for such a thorough and insightful review of our paper. We also appreciate the reviewer's strong support for the paper. Below are our responses to your questions. Please do not hesitate to raise further questions.
>
> ----
> **Response to Weakness 1 "SYSTEM COMPLEXITY" and Q1:** We have analyzed the wall-clock run-time of each module in Table 1 in supplementary material. The full pipeline processes a LingoQA video in <4.5 minutes with unoptimized Python (Table 1 in supplementary material), suitable for offline post-hoc video analysis. Further (based on Reviewer zz3N's suggestion) we note the following. On a single A6000 GPU with 48 GB memory, the end-to-end runtime of ı̇Finder for video inputs of 5 seconds, 10 seconds, and 30 seconds (at 10 FPS with image size 640x360) is approximately 2.33 minutes, 3.85 minutes, and 18.75 minutes, respectively.
>
> ----
> **Response to Weakness 2 "RELIANCE ON RULE-BASED STRUCTURE":** We agree that ı̇Finder targets the driving domain. However, we highlight the following that would enable generalization:
>    - Our representation is auto-generated via standard pretrained models (e.g., OWL-V2, SAM, CenterTrack) with no task-specific fine-tuning (Sec. 3.1). This decoupled design generalizes to any domain where object-level cues (position, orientation, interaction) are relevant.
>    - The robustness under visual noise is empirically evaluated on MM-AU. Even after filtering out >97% of object detections (τ = 0.8), accuracy remains 58.27%, surpassing all baselines including VideoLLaMA2 (52.89%) (Table 7). Our symbolic reasoning retains reliability under perceptual degradation.
>    - In Table 5, we also show that cross-condition generalization is strong. In adversarial settings (rain, snow, night), ı̇Finder outperforms the best V-VLM by 10–20% across all categories (Table 5), despite no domain adaptation. This suggests high resilience even under covariate shift.
>
> While domain transfer is not the paper’s focus, we believe ı̇Finder offers a template for symbolic grounding that can be instantiated in new domains via different pretrained perception modules. We will include this as a direction for future work.
>
> ----
> **Response to Weakness 3 "UNCLEAR PEER CONTRIBUTION":** Note that iFinder improves the peer V-VLM by correcting its initial response with the proposed representational abstraction of vision inputs that enables symbolic reasoning over dynamic scenes using LLMs. We provided the impact of different peers within iFinder on LingoQA dataset in Table below. It can be observed that iFinder can improve enhance the overall performance irrespective of  the peer V-VLM involved.
> |                      | Acc(%) |
> |:--------------------:|:------:|
> | VideoLLama2          | 36.00  |
> | **Ours** (w/ VideoLLama2) | **41.20**  |
> | VideoChat2           | 41.20  |
> | **Ours** (w/ VideoChat2)  | **44.20**  |
>
> ----
> **Response to [Additional Comments] 1 & Q4:**
>  - We completely agree with the Reviewer that adding a discussion as well as evaluating iFinder on works like reviewer references [1, 2, 3] would demonstrate interesting insights and strengthen iFinder's position. We will add a discussion on these works in the related works section in the revised version of the manuscript.
>  - In response to the reviewer’s suggestion to evaluate on the "Coool" benchmark [1], we explored the corresponding GitHub repository but found that ground truth annotations were not publicly available as of July 31, 2025. Despite this limitation, we conducted a qualitative analysis using an example to demonstrate the strengths of iFinder in Table below.
> | video name     | Github Baseline response (manually reformatted for better readability in the table)                                                                                                                                                                                                                                                                                                                                                                                                                                                                                                   |  iFinder response                                                                                                                                                                                                       | GT (visually observed)                                                               |
> |----------------|-------------------------------------------------------------------------------------------------------------------------------------------------------------------------------------------------------------------------------------------------------------------------------------------------------------------------------------------------------------------------------------------------------------------------------------------------------------------------------------------------------------------------------------------------------------------------------|-------------------------------------------------------------------------------------------------------------------------------------------------------------------------------------------------------------------------|------------------------------------------------------------------------------------|
> | video_0002.mp4 | Frame 2-109, 151-360: there is a white and black cat sitting on a table  focus and ( gaussian blur )  144p  may 1 0  white fog  extreme panoramic  zoom  see  zoom out  full width  teaser  dribbble 8k  8 k matte painting  may 6 8  high resolution image  untextured  without glasses  - signature Frame 110-150: there is a deer that is running down the street  discord profile picture  low quality footage  his legs spread apart  cryptid  captures emotion and movement  extremely close shot  half dog  loosely cropped  youtube thumbnail  moose  dynamic closeup | Animal in same lane as ego vehicle (frames 102-108, 116-119). Animal changing lanes unpredictably (frames 120-125). Car approaching ego vehicle (frames 151-152, 157-159). Debris on wet road (surrounding_info field). | An animal ran from the right side and crossed in front of the ego car to the left. |
>  - Note that we evaluated iFinder under adversarial conditions (Table 5) in "Foggy", "Rainy", "Snowy" and "Night" conditions where it outperforms baselines (VideoLLama2 and VideoLLaVA) by (8.33%, 20.69%, 17.61%, 13.7%) and (16.67%, 24.14%, 15.09%, 17.35%), respectively. The improved accuracy and robustness validate the extremely significant value of leveraging vision-language grounding offered by iFinder.
>
> ----
>  **Response to [Additional Comments] 2:** We agree with the reviewer's suggestion. We will add the failure cases of iFinder in the revised manuscript. Please see below two examples from the Nexar dataset, marked by their respective video names (.mp4 files) where iFinder did not produce the expected response. We provide an analysis on possible reasons for these examples as well.
>    - “01031”: This dash cam video shows a vehicle cutting in from the left side of the ego car to the front. Since there is no obvious steering or vibration caused by a collision, it is even difficult for humans to determine whether a collision actually occurred.
>    - “00970”: This dash cam video shows a vehicle suddenly braking in front of the ego car. Although it is very close to the ego car, the video also shows the ego car braking, so it is not clear from the video whether an actual collision occurred.
>
>
> ----
>  **Response to [Additional Comments] 3 & Limitations 4 & Q5:** Based on reviewer's suggestion, we ran the iFinder module end-to-end with different random seeds on LingoQA dataset and provide the analysis below. Also note that we used fixed temperature for the LLM.
>
> |           | Acc(%) |
> |:------:|:------:|
> | Run 1 | 43.80  |
> | Run 2 | 44.20  |
> | Run 3 | 44.20  |
>
> ----
>  **Response to comments under [Grammar & Typos & Formating]:** Thank you for pointing this out. We agree with the reviewer and will fix this in the final version of the manuscript.
>
> ----
> **Response to Q2:** Yes, we plan to extend iFinder toward semi-real-time scenarios (e.g. using quantized version of each module). The major (and exciting) challenge on this will be to maintain the performance of the adopted modules in their quantized form.
>
> ----
> **Response to Q3 & Limitations 2:** Yes, iFinder’s pipeline is adaptable to domains beyond driving. This is because its schema—temporal object-centric data—is domain-structured, not domain-locked. Its modular design allows substitution of perception modules for other domains (e.g., indoor robotics), and the JSON format can accommodate arbitrary scene cues. Our ablations (Table 6) show performance is robust to component removal, indicating generalizable reasoning.
>
> ----

---

> > ### Comment · Reviewer_XuKK · 2025-08-01
> >
> > Thank you for your detailed rebuttal. I appreciate your clarification and response to my comments.
> >
> > My review and questions were intended to help strengthen the paper, and I am glad to see that the planned improvements address my main "concerns". Given my already positive assessment, I will maintain my current rating.

---

> > > ### Author Response · Authors · 2025-08-01
> > >
> > > Hi Reviewer XuKK,
> > >
> > > Thank you for the support and your strong positive rating. We will incorporate all comments in the revised version of the paper.
> > >
> > > Best Wishes, \
> > > Authors

---

### Official Review · Reviewer_zz3N · 2025-07-08

**Clarity:** 3
**Significance:** 2
**Originality:** 3
**Rating:** 4
**Confidence:** 3

**Summary:**

The paper introduces ı̇Finder, a training-free pipeline for post-hoc reasoning over dash-cam videos. It decouples perception from language reasoning by (i) extracting driving-specific cues—object pose, lane location, distances, ego motion, 3D orientation and scene context—using off-the-shelf vision models, (ii) packaging those cues into a hierarchical JSON structure, and (iii) feeding the structure, together with a peer VLM’s first-pass answer, to a GPT-4-class LLM via a three-block prompt (key explanations, step instructions, peer instruction). On four zero-shot driving benchmarks (MM-AU, SUTD-TrafficQA, LingoQA, Nexar), experimental results show that ı̇Finder outperforms both general-purpose Video-VLMs and driving-specialised models.

**Questions:**

1. What is the end-to-end runtime of ı̇Finder per 5s/10s/30s/60s video? A breakdown would clarify deployment feasibility.

2. If stronger/weaker object trackers or depth estimators are substituted, do the gains persist? Ablation study can be conducted by using weaker object trackers and check their performance with iFinder.

3. Discuss two weaknesses.

**Ethical Concerns:**

["NO or VERY MINOR ethics concerns only"]

**Final Justification:**

Thanks for the author's reply. Although the latency is still a issue for the proposed model, I really appreciate comparison and detailed explanation of the answers. I will remain my positive score.

**Limitations:**

They discussed it in the paper.

**Paper Formatting Concerns:**

Looks good to me.

**Quality:**

3

**Strengths And Weaknesses:**

**Strengths:**

- It has solid engineering for all vision modules and connect them successfully with GPT-4o.

- The paper has rigorous zero-shot evaluation on 4 public ego-view driving benchmarks. Each of them contain problems that current VLMs can not resolve alone.

- The paper combines peer-LLM self-correction with structured visual grounding—a fresh twist on “LLM-agent” ideas for autonomous driving.

**Weaknesses**

- Real-time latency/cost of running eight vision modules is not analysed. This might be a big problem if one or more module take a long time during inference. Besides, It relies exclusively on existing pretrained components. What if some module make mistakes? This is not discuss in the ablation study.

- It seems that gains come largely from adding extra model-derived priors (3D orientation, depth) rather than novel learning; may be less compelling once future VideoVLMs close the gap.

---

> ### Author Rebuttal · Authors · 2025-07-31
>
> Thank you for the thoughtful comments and support for our paper. Below are our responses to your questions. Please do not hesitate to raise further questions.
>
> ----
> - Response to  ***Real-time latency/cost of running eight vision modules is not analysed. This might be a big problem if one or more module take a long time during inference.***
>     - We have analyzed the wall-clock run-time of each module in Table 1 in supplementary material. The full pipeline processes a LingoQA video in <4.5 minutes with unoptimized Python (Table 1 in supplementary material), suitable for offline post-hoc video analysis.
>
> ----
>
> - Response to ***Besides, It relies exclusively on existing pretrained components. What if some module make mistakes? This is not discuss in the ablation study.***
>     - We analyzed the impact of error propagation (or effect of compounding errors when the initial module, in our case the object detector, fails to detect objects in the scene) in Table 7 on MM-AU dataset. For example, to imitate this detection failure, we remove >97% of object detections (τ = 0.8) from the JSON files and observed that the overall accuracy still remains 58.27%, surpassing all baselines including VideoLLaMA2 (52.89%) (Table 7). This shows that our symbolic reasoning retains reliability under perceptual degradation. Further, Table 7 also shows that iFinder demonstrates graceful degradation and limited error propagation. Please see L326-333 for a detailed discussion.
>
> ----
>
> - Response to ***It seems that gains come largely from adding extra model-derived priors (3D orientation, depth) rather than novel learning; may be less compelling once future VideoVLMs close the gap.***
>     - We respectfully clarify that the gains in ı̇Finder result not from simply adding priors, but from a novel structured grounding framework that integrates visual cues into LLM reasoning via symbolic representations. Three quantitative findings support this:
>         - Removing 3D orientation reduces MM-AU accuracy from 63.39% → 58.83%, but this is still 5.94% above VideoLLaMA2 (52.89%), showing priors are necessary but not sufficient (Table 6). The largest drop occurs only when structured scene understanding is removed (→ 57.81%).
>         - End-to-end VLMs do not close the gap despite strong priors. VideoLLaMA2 and DriveMM have implicit access to visual cues but still lag by 10.5% and 39.17%, respectively, on MM-AU (Table 1). Structured reasoning over priors is critical, not their mere presence.
>         - Removing only “Key Explanation” in our prompt drops accuracy by 4.66%, more than removing any single visual input (Table 6). This suggests performance stems from reasoning over structure, not data quantity.
>
>     Our method is modular, zero-shot, and interpretable, offering complementary strengths to future end-to-end VLMs. We hope this clarifies that ı̇Finder is not a wrapper over priors, but a distinct, architectural contribution to structured LLM grounding.
>
> ----
>
>  - Response to ***What is the end-to-end runtime of ı̇Finder per 5s/10s/30s/60s video? A breakdown would clarify deployment feasibility.***
>     - On a single A6000 GPU with 48 GB memory, the end-to-end runtime of ı̇Finder for video inputs of 5 seconds, 10 seconds, and 30 seconds (at 10 FPS with image size 640x360) is approximately 2.33 minutes, 3.85 minutes, and 18.75 minutes, respectively.
>
> ----
>
> - Response to ***If stronger/weaker object trackers or depth estimators are substituted, do the gains persist? Ablation study can be conducted by using weaker object trackers and check their performance with iFinder.***
>     - Thanks for the suggestion. We  replaced the tracker and distance estimation modules with their weaker counterparts and analyzed the performance of iFinder on MM-AU dataset. We observed that replacing the tracker with a weaker tracker (simple IoU based tracker vs ByteTracker) drops performance slightly to 61.70%, indicating that tracking quality contributes moderately to system accuracy. Similarly, using a weaker depth model (Base vs Large) causes a minor drop to 62.78%.
>
> ----
>
> - Response to ***Discuss two weaknesses.***
>     - We provide two observable weaknesses w.r.t. post-hoc analysis that future works can build upon.
>         - The goal of this work is to demonstrate how vision-based modules can effectively ground large language models (LLMs) for robust reasoning, without relying on external sensors like LiDAR or GPS. By focusing solely on visual input, iFinder highlights the potential of vision as a standalone modality for scene understanding and decision-making. While current implementation does not integrate LiDAR or GPS data, this design choice underscores the strength of vision-driven grounding and leaves open the possibility for future multi-modal extensions to further enhance performance in complex scenarios.
>         - Being entirely training-free, iFinder cannot self-correct based on prior mistakes or user feedback. Such adaptability can be built-in by adding in-context examples or LLM fine-tuning.
>
> ----

---

> > ### Author Response · Authors · 2025-08-04
> >
> > Hi Reviewer zz3N,
> >
> > Thank you again for reviewing our work and insightful comments. We wanted to check if our responses have clarified your questions. Your support is extremely valuable to us.
> >
> > Best wishes, \
> > Authors

---

> > ### Comment · Reviewer_zz3N · 2025-08-06
> >
> > Thanks for the author's reply.
> > Although the latency is still a issue for the proposed model, I really appreciate comparison and detailed explanation of the answers. I will remain my positive score. Good luck.

---

> > > ### Author Response · Authors · 2025-08-06
> > >
> > > Hi Reviewer zz3N,
> > >
> > > Thank you for your thoughtful feedback and for taking the time to review our work. We **sincerely appreciate** your positive assessment and acknowledge that latency remains an area for improvement. Regarding your question on real-time operation: iFinder can be adapted for real-time use depending on the available compute. On limited hardware, more efficient model variants can be substituted to reduce latency. With sufficient resources (e.g., multiple high-end GPUs), the system can operate through hosted APIs with parallel execution to enable real-time performance.
> > >
> > > Your comments have been very helpful, and we will incorporate all comments in the revised version of the paper.
> > >
> > > Best wishes, \
> > > Authors

---

### Note · Authors · 2025-08-11

Dear AC, SACs, PCs and Reviewers,

In our paper, we introduce iFinder, a vision-language system designed for post-hoc dash-cam video analysis. Our core idea is to show that introducing a representational abstraction of videos - effectively decoupling low-level perception from high-level reasoning - can enable symbolic reasoning over dynamic scenes using LLMs, thereby allowing the language model to be grounded in domain-specific tasks. iFinder leverages pre-trained components (e.g., object detection, depth estimation, 3D pose) to extract these representational abstraction of videos, and integrates them into a coherent, temporally structured, LLM-compatible JSON schema for the subsequent reasoning. Although the vision perception components are off-the-shelf, iFinder unifies outputs from multiple modalities and resolutions (2D, 3D, semantic, temporal) into a schema that preserves per-frame and global consistency, while transforming raw signals (e.g., camera pose, lane estimation) into interpretable abstractions (e.g., “left turn,” “cutting lanes”) to enable higher-level symbolic reasoning by LLMs. Our experiments show that unlike baselines [14–16, 21] that provide incorrect or generic explanations, iFinder correctly leverages the JSON schema, allowing the LLM to draw grounded, and reliable conclusions.

We are happy that all reviewers recognized multiple strengths of our paper:

 - Rev. zz3N: a fresh twist on “LLM-agent” ideas for autonomous driving; novel peer-LLM self-correction with structured visual grounding, validated on 4 ego-view driving benchmarks.
 - Rev. XuKK: clear, well-written, and well-structured with excellent implementation details for reproducibility; modular, training-free design integrating domain knowledge; highly relevant problem in vision-language understanding for not just autonomous systems
 - Rev. XWwS: well-written, training-free method with good interpretability; competitive results across multiple benchmarks.
 - Rev. qGHL: structured symbolic cues enable interpretable reasoning in complex driving scenarios; sufficient empirical analysis showing the framework significantly outperforming both general-purpose and driving-specific VLMs.

We are pleased that Reviewers XWwS and qGHL raised their ratings positively from 3 to 4, while Reviewers zz3N and XuKK maintained positive ratings of 4 and 6 (post-rebuttal). We will incorporate all comments in the revised paper.

Thanks for your time!

Best wishes, \
Authors

---

### Decision · Program_Chairs · 2025-09-17

**Decision:**

Accept (poster)

**Comment:**

The paper presented iFinder, a novel, modular, training-free framework for dash-cam video reasoning by LLM. iFinder is a structured grounding framework that separates low-level perception from high-level reasoning to enhance LLM-based video understanding. This work designs a modular and training-free pipeline that combines multiple pretrained perception modules to generate interpretable inputs for LLMs. The experiments results show strong zero-shot performance on multiple driving datasets with thorough ablations on prompt structure, perception module substitution, and adversarial conditions. The reviewers also pointed out some weaknesses, including the limited novelty in model components, efficiency issue due to the multi-module pipeline,
During the rebuttal phase, the authors effectively addressed the reviewers' concerns about latency, ablation coverage, and adaptability. The final ratings include a Strong Accept, and 3 Borderline Accepts, with 2 review ratings raised from initial Borderline Rejects). The authors' rebuttal improved the perception of the paper. The authors provided clarifications on latency, ablation studies, peer model contribution, and robustness to perception noise. The two initially most critical reviewers (XWwS and qGHL) explicitly noting that the authors' rebuttals addressed their concerns, justifying their final score adjustments.
In summary, this paper is technically sound, addresses an important problem in a principled manner, and shows promising experimental results on four benchmarking datasets. This paper provides excellent contributions to grounded vision-language reasoning. The AC agrees with the reviewers' suggestions and recommends accepting the paper.